

**Direct radiative effects of intense Mediterranean desert dust outbreaks**
Antonis Gkikas[1,2], Vincenzo Obiso[2], Carlos Pérez García-Pando[2], Oriol Jorba[2], Nikos Hatzianastassiou[3],
Lluis Vendrell[2], Sara Basart[2], Santiago Gassó[4] and José Maria Baldasano[2,4]
[1]Institute for Astronomy, Astrophysics, Space Applications and Remote Sensing, National Observatory of Athens, Athens,
15236, Greece
[2]Earth Sciences Department, Barcelona Supercomputing Center, Barcelona, Spain
[3]Laboratory of Meteorology, Department of Physics, University of Ioannina, Ioannina, Greece
[4]Environmental Modelling Laboratory, Technical University of Catalonia, Barcelona, Spain
Corresponding author: Antonis Gkikas (agkikas@noa.gr)
**Abstract**

17       The direct radiative effect (DRE) of 20 intense and widespread dust outbreaks that affected the broader

Mediterranean basin during the period March 2000 – February 2013, has been calculated with the
regional NMMB-MONARCH model. The DREs have been calculated based on short-term simulations
(84 hours) for a domain covering the Sahara and most part of the European continent. At midday, desert
dust outbreaks induce locally a NET (shortwave plus longwave) strong atmospheric warming ($DRE_{ATM}$
values up to 285 Wm$^{-2}$), a strong surface cooling ($DRE_{NETSURF}$ values down to -337 Wm$^{-2}$) whereas they
strongly reduce the downward radiation at the ground ($DRE_{SURF}$ values down to -589 Wm$^{-2}$). During
nighttime, reverse effects of smaller magnitude are found. At the top of the atmosphere (TOA), positive
(planetary warming) DREs up to 85 Wm$^{-2}$ are found over highly reflective surfaces while negative
(planetary cooling) DREs down to -184 Wm$^{-2}$ are computed over dark surfaces at noon. Desert dust
outbreaks significantly affect the regional radiation budget, with regional clear-sky NET DRE values
ranging from -13.9 to 2.6 Wm$^{-2}$, from -43.6 to 4 Wm$^{-2}$, from -26.3 to 3.9 Wm$^{-2}$ and from -3.7 to 28 Wm$^{-2}$
$^{2}$ for TOA, SURF, NETSURF and ATM, respectively. Although the shortwave (SW) DREs are larger
than the longwave (LW) ones, the latter are comparable or even larger at TOA, particularly over the
Sahara at midday. As a response to the strong surface cooling during daytime, dust outbreaks cause a
reduction of the regional sensible and latent heat fluxes by up to 45 Wm$^{-2}$ and 4 Wm$^{-2}$, respectively,
averaged over land areas of the simulation domain. Dust outbreaks reduce the temperature at 2 meters
by up to 4 K during day, whereas a reverse tendency of similar magnitude is found during night.
Depending on the vertical distribution of dust loads and time, mineral particles heat (cool) the atmosphere





by up to 0.9 K (0.8 K) during daytime (nighttime) within atmospheric dust layers. Beneath and above the
dust clouds, mineral particles cool (warm) the atmosphere by up to 1.3 K (1.2 K) at noon (night). When
dust radiative effects are taken into account in numerical simulations, the total emitted dust and dust
AOD, computed on a regional mean basis, are decreased (negative feedback) by 19.5% and 6.9%. The
consideration of dust radiative effects in numerical simulations improves the model predictive skills.
More specifically, it reduces the model positive and negative biases for the downward surface SW and
LW radiation, respectively, with respect to Baseline Surface Radiation Network (BSRN) measurements.
In addition, they also reduce the model near-surface (at 2 meters) nocturnal cold biases by up to 0.5 K
(regional averages), as well as the model warm biases at 950 and 700 hPa, where the dust concentration
is maximized, by up to 0.4 K.

## 1.  Introduction

Dust aerosols through their interaction with the incoming solar (shortwave, SW) and the outgoing
terrestrial (longwave, LW) radiation, perturb the radiation budget of the Earth-Atmosphere system and
redistribute the energy therein. The induced perturbation of the radiation fields by dust particles, the so-
called dust radiative effect, takes place through three processes of increasing complexity affecting the
energy budgets at the surface, into the atmosphere and at the top of the atmosphere (TOA). The first one,
known as direct radiative effect (DRE) and referred as REari (aerosol-radiation interactions) in the latest
report of the Intergovernmental Panel on Climate Change (IPCC, Boucher et al., 2013), is caused by the
absorption and scattering of the SW radiation (Sokolik et al., 2001) and the absorption and re-emission
of the LW radiation by mineral particles (Heinold et al., 2008). Due to the perturbation of the radiation
fields by dust aerosols, the energy budget both at the surface and into the atmosphere is modified and the
signal of these impacts is evident in atmospheric stability/instability conditions associated with cloud
development and precipitation. These rapid adjustments, which have been earlier referred as semi-direct
effects (Hansen et al., 1997), are induced by the dust REari on surface energy budget and atmospheric
profile (Boucher et al., 2013) contributing to the Effective Radiative Forcing (ERFari). Moreover, dust
aerosols due to their ability to serve as cloud condensation nuclei (CCN) and ice nuclei (IN), modify the
physical (Twomey, 1974; Albrecht, 1989) and optical properties of clouds (Pincus and Baker, 1994),
which consist the major regulators of the Earth-Atmosphere system's radiation budget (Lohmann and
Feicher, 2005). Through this chain of complex processes, it is described the indirect impact of mineral
particles on the radiation and compared to the other two dust radiative effects (direct and semi-direct) is
characterized by even larger uncertainties. In the latest IPCC report (IPCC, 2013), the formerly known



as indirect effects have been renamed to Effective Radiative Forcing (ERFaci) including the modification

of radiation by clouds, attributed to aerosol-cloud interactions (aci), as well as the subsequent changes

(rapid adjustments) of clouds' physical/microphysical/optical properties (Boucher et al., 2013).

Several studies have been conducted aiming at estimating the dust direct/semi-direct (e.g. Pérez et al.,

2006; Helmert et al., 2007; Zhao et al., 2010; Nabat et al., 2015a) and indirect effects (e.g. Sassen et al.,

2003; Seigel et al., 2013). Specifically, numerous studies have been carried out either by means of

numerical modelling (e.g. Solmon et al., 2012; Woodage and Woodward, 2014) or through the synergy

of observations and radiative transfer codes (Di Sarra et al., 2011; Valenzuela et al., 2012) or solely based

on aerosol observations (e.g. Yang et al., 2009; Zhang et al., 2016) and their findings either referred to

extended (e.g. Spyrou et al., 2013) or limited time periods (e.g. Nabat et al., 2015b) or to specific desert

dust outbreaks (e.g. Pérez et al., 2006; Santese et al., 2010; Stanelle et al., 2010). The investigation of

dust radiative effects is a scientific issue of great concern since it is documented that mineral particles,

through their interaction with the radiation, can affect atmospheric processes from short (weather) to long

(climate) temporal scales. To this aim, many research efforts were dedicated to the investigation of dust

impacts on the convective activity (Mallet et al., 2009), sea surface temperature (Foltz and McPhaden,

2008), hydrological cycle (Miller et al., 2004b), hurricanes (Bretl et al., 2015), boundary layer dynamics

(Heinold et al., 2008) and monsoons (Solmon et al., 2008; Vinoj et al., 2014).

The direct impact of dust aerosols is expressed by the sign and the magnitude of the DRE values,

which are defined as the anomalies (perturbation) of the radiation attributed to dust-radiation direct

interaction, considering as a reference (control) an atmospheric state where mineral particles are not a

radiatively active substance. Based on this, negative and positive DREs indicate a cooling (loss of energy)

and a warming effect (gain of energy), respectively. Nevertheless, the sign of the DREs varies between

the SW and LW spectrum (Osborne et al., 2011) as well as within the Earth-Atmosphere system. More

specifically, due to the attenuation (through scattering and absorption) of the SW radiation, dust aerosols

warm the atmosphere and cool the surface (Huang et al., 2014), while reverse tendencies are revealed at

longer wavelengths attributed to the absorption and re-emission of LW radiation by the mineral particles

(Sicard et al., 2014a). Between the two spectrum ranges, the SW DREs are larger compared to the LW

ones, in absolute terms, explaining thus their predominance when the corresponding calculations are

made for the NET (SW+LW) radiation (e.g. Pérez et al., 2006; Zhu et al., 2007; Woodage et al., 2014).

The perturbations of the radiation budget at the surface and into the atmosphere determine the DRE at

TOA (e.g. Kumar et al., 2014), which indicates the increase (planetary cooling) or the decrease (planetary



warming) of the outgoing radiation from the Earth-Atmosphere system and is relevant to dust climatic
effects (Christopher and Jones, 2007).

The scientific importance of investigating the dust direct impacts on radiation has been notified in

previous studies where it was shown that the consideration of the dust-radiation interactions may improve
the forecasting ability of weather models (Pérez et al., 2006) and can reduce the observed biases of the
LW radiation at TOA between models and satellite retrievals (Haywood et al., 2005). The dust direct
impacts are highly variable both in space (e.g. Zhao et al., 2010) and time (e.g. Osipov et al., 2015)
attributed to several parameters related either to dust aerosols' physical and optical properties or to
external factors (e.g. surface type), which determine both the sign and the magnitude of the DREs (Liao
and Seinfeld, 1998). One of the most important factor is the composition of mineral particles determining
the spectral variation of the refractive index (Müller et al., 2009; Petzold et al., 2009) and subsequently
their absorption efficiency (Mallet et al., 2009), which are both critical in radiation transfer studies, and
are also dependent on the mixing state (either external or internal) of dust aerosols (Scarnato et al., 2015).
Under clear skies, apart from mineral particles' optical properties, the shape (Wang et al., 2013a), the
emitted dust size distribution (Mahowald et al., 2014), the surface albedo (Tegen et al., 2010) as well as
the vertical distribution of dust aerosols (Mishra et al., 2015) have been recognized as determinant factors
for the DRE calculation. On the contrary, when clouds are present, the position of dust layers with regards
to clouds defines the sign and the magnitude of DREs at TOA (Yorks et al., 2009; Meyer et al., 2013;
Choobari et al., 2014; Zhang et al., 2014).

The dust radiative effects become important under specific conditions of very high concentrations, so-

called events or episodes or outbreaks. Such episodes occur frequently over the broader Mediterranean
basin (Gkikas et al., 2013), due to its vicinity to the world's major dust sources situated across the
northern Africa (Sahara) and Middle East deserts (Ginoux et al., 2012). Dust particles are mobilized over
these areas by strong winds (Schepanski et al., 2009) being uplifted to the free troposphere due to strong
convection in the boundary layer (Cuesta et al., 2009) and are transported towards the Mediterranean due
to the prevailing atmospheric (synoptic) circulation (Gkikas et al., 2015). Under these conditions, dust
particles over the Mediterranean are recorded at very high concentrations as it has been confirmed either
by satellite (e.g. Moulin et al., 1998; Guerrero-Rascado et al., 2009; Rémy et al., 2015) and ground
retrievals (e.g. Kubilay et al., 2003; Toledano et al., 2007) or by surface $PM_{10}$ measurements (e.g.
Rodríguez et al., 2001; Querol et al., 2009; Pey et al., 2013).

Among the different aerosol types that co-exist in the Mediterranean (Lelieveld et al., 2002; Basart et

al., 2009), dust is the one causing the greatest perturbation of the SW and LW radiation, especially during



desert dust outbreaks (e.g. Di Sarra et al., 2008; Di Biagio et al., 2010). Thus, a number of studies focused
on Mediterranean dust outbreaks' impacts on the SW (Meloni et al., 2004; Gómez-Amo et al., 2011;
Antón et al., 2012; Di Sarra et al., 2013; Obregón et al., 2015), LW (Antón et al., 2014; Sicard et al.,
2014a) and NET (Di Sarra et al., 2011; Romano et al., 2016) radiation. However, the obtained results
were representative at a local scale and considering the high spatial variability of desert dust outbreaks,
the optimum solution of assessing in a comprehensive way their impacts on weather and climate is
provided by atmospheric-dust models. To this aim, the induced DREs by the Mediterranean desert dust
outbreaks have been analyzed through short-term numerical simulations (Pérez et al., 2006; Santese et
al., 2010; Remy et al., 2015), revealing strong perturbations of the energy budget within the Earth-
Atmosphere system, which in turn affect atmospheric processes. Moreover, similar studies have been
conducted either at a seasonal (Nabat et al., 2015a) and annual scale (Nabat et al., 2012) or for extended
time periods (Spyrou et al., 2013; Nabat et al., 2015b) pointing out the key role of desert dust aerosols in
the Mediterranean climate.
The overarching goals of the present study are: (i) the assessment of the short-term direct radiative
effects (DREs) on the Earth-Atmosphere system's radiation budget, induced by intense Mediterranean
desert dust outbreaks, based on regional model simulations, (ii) the assessment of the associated impacts
on temperature and sensible/latent heat fluxes, (iii) the investigation of possible feedbacks on dust AOD
and dust emission and (iv) the assessment of the model's predictive skills, in terms of reproducing
temperature and radiation fields, when dust-radiation interactions are taken into account in numerical
simulations. To this aim, 20 intense and widespread desert dust outbreaks that affected the broader area
of the Mediterranean basin, over the period March 2000 – February 2013, have been identified based on
an objective and dynamic satellite algorithm (Section 2). It must be highlighted that through the
consideration of a large dataset of desert dust outbreaks is ensured the robustness of our findings,
providing thus the opportunity to have a clear view of dust outbreaks' impacts on radiation as well as
about the associated impacts on meteorological variables (e.g. temperature). For each dust outbreak,
through short-term (84 h) numerical simulations of the regional NMMB-MONARCH model (Section 3),
the DREs are calculated at TOA, surface and into the atmosphere, both at grid point (geographical
distributions) and regional scale level (Section 5.2), for the SW, LW and NET (SW+LW) radiation. In
addition, are examined the impacts of the Mediterranean desert dust outbreaks on the sensible/latent heat
fluxes (Section 5.3) and on the surface temperature (Section 5.4) as well as the potential feedbacks on
dust AOD and dust emissions (Section 5.5). The last part of the study (Sections 5.6 and 5.7) investigates
the potential improvement of the model's forecasting ability in terms of reproducing the temperature and



radiation fields when dust-radiation interactions are included in numerical simulations. A summary is
made and conclusions are drawn in Section 6.

**2. Selection of desert dust outbreaks**

In the present study, 20 intense and widespread desert dust outbreaks that affected the broader area of
the Mediterranean basin, over the period March 2000 – February 2013, are analyzed. The studied desert
dust outbreaks have been identified using an objective and dynamic satellite algorithm introduced in
Gkikas et al. (2013) and further developed in Gkikas et al. (2016). The algorithm utilizes daily 1° x 1°
latitude-longitude resolution satellite retrievals, derived from MODerate resolution Imaging
Spectroradiometer (MODIS; Remer et al., 2005), Total Ozone Mapping Spectrometer (TOMS; Torres et
al., 1998) and Ozone Monitoring Instrument (OMI; Torres et al., 2007) observations. The MODIS-Terra
(Collection 051) aerosol optical depth at 550nm ($AOD_{550nm}$), Ångström exponent ($\alpha$), fine fraction ($FF$)
and effective radius ($r_{eff}$, available only over sea) products are used in the algorithm along with EP-TOMS
and OMI-Aura Aerosol Index ($AI$). Using these products, the algorithm takes into account information
regarding aerosols' load ($AOD$), size ($FF$, $a$ and $r_{eff}$) and absorbing/scattering ability ($AI$) which is
necessary for the identification of dust.
Only a brief discussion of the algorithm operation is given here, whereas a detailed description is
provided in Gkikas et al. (2013). The satellite algorithm is applied to each individual 1° x 1° grid cell of
the Mediterranean Satellite Domain (29° N - 47° N and 11º W - 39º E, MSD, red rectangular in Figure
1) during the period March 2000 – February 2013. For each grid cell, from the series (2000-2013) of
daily $AOD_{550nm}$ values, the mean (*Mean*) and the associated standard deviation (*Std*) of $AOD_{550nm}$ are
calculated. Based on these two primary statistics, two threshold (or cut-off) levels being equal to
*Mean+2\*Std* and *Mean+4\*Std,* are defined. By comparing each daily AOD value to the two thresholds,
the algorithm determines whether an aerosol episode (or event) occurs over an 1° x 1° grid cell in that
day or not, and labels it as strong or extreme, depending on which AOD threshold is exceeded (lower or
higher). Thereby, the term "aerosol episode" refers to pixel-level episodic (extremely high loading)
aerosol conditions and it is used with this meaning henceforth. Subsequently, in order to characterize the
identified pixel-level episodes as desert dust (DD) ones, appropriate thresholds for $\alpha$, $FF$, $r_{eff}$ and $AI$ are
used, based on existing knowledge about relevant physical properties (size and absorbing/scattering
ability) of dust. According to the algorithm, a strong or extreme pixel-level DD episode occurs if $\alpha \leq 0.7$,
$FF \leq 0.4$, $r_{eff} > 0.6$ μm and $AI > 1$ (conditions should be met simultaneously).



Based on the satellite algorithm's outputs, for each day of the study period it is calculated the total
number of grid cells over which a strong or an extreme DD episode has taken place. Subsequently, from
the overall series of 4748 days over the study period, are kept only those in which at least 30 grid cells
with a DD episode (either strong or extreme) have been recorded. This criterion was first adopted by
Gkikas et al. (2015), who analyzed the atmospheric circulation evolution patterns favoring the occurrence
of dust outbreaks over the broader Mediterranean basin, in order to keep and study the most extensive
ones (in terms of the number of pixel-level DD episodes). In a next step, the days satisfying the defined
criterion (i.e. days where at least 30 pixel-level DD episodes have been occurred) are ranked based on
their regional MODIS-Terra AODs averaged over the "dust episodic" pixels within the geographical
limits of the MSD. If two or more consecutive days are satisfying the defined criteria, then the day with
the maximum number of DD episodes is selected. The final dataset consists of 20 intense Mediterranean
desert dust outbreaks listed in a chronological order in Table 1.
The majority of the selected desert dust outbreaks (55 % or 11 out of 20) took place in spring (March-
April-May) when massive dust loads originating in the Sahara Desert are transported towards the central
and eastern parts of the Mediterranean (Gkikas et al., 2013; Pey et al., 2013). Four widespread desert
dust outbreaks affected mainly the western sector of the MSD in summer (July, August), while five dust
outbreaks were recorded across the central and eastern parts of the basin in winter (January, February).
Among the selected cases, the number of pixel-level DD episodes in the MSD varies from 30 (28 July
2005, western-central Mediterranean) to 85 (31 July 2001, western Mediterranean), whereas their
intensity (in terms of AOD at 550nm) ranges from 0.74 (31 July 2001) to 2.96 (2 March 2005), being in
general higher in winter while moderate-to-high intensities are recorded in spring. Based on the
information in Table 1, the selected study cases correspond to widespread and intense dust outbreaks that
occurred in various parts of the Mediterranean, and therefore they are representative and appropriate for
further studying their radiative effects. The occurrence of intense desert dust outbreaks during the first
half of the year is favored either by the predominance of intense low pressure systems across the
Mediterranean basin (Varga et al., 2014; Gkikas et al., 2015) or by their eastwards shift (Saharan
depressions) across the northern coasts of Africa (Alpert and Ziv, 1989). In both seasons, dust transport
from the northern Africa deserts towards the Mediterranean is induced by the prevailing southerly or
southwesterly airflow (Barkan et al., 2005; Meloni et al., 2008). Some of the identified desert dust
outbreaks here, have been also analyzed in previous studies related to particulate matter levels
(Kanakidou et al., 2010), chemical speciation (Theodosi et al., 2010), dust layers' vertical structure
(Amiridis et al., 2009; DeSouza-Machado et al., 2010), dust radiative effects (Di Sarra et al., 2011), dust



modelling (Carnevale et al., 2012) and prevailing synoptic conditions favoring the occurrence of dust
events (Nastos, 2012).
**3. Model description**
In the present section, the main features of the meteorological driver (Section 3.1.1) and the dust
module (Section 3.1.2) used in the regional NMMB-MONARCH (Multiscale Online Nonhydrostatic
AtmospheRe CHemistry) model, previously known as NMMB/BSC-Dust, are described. The version
(v1.0) of the NMMB-MONARCH model used here contributes to different model inter-comparisons like
the International Cooperative for Aerosol Prediction (ICAP) initiative and the Sand and Dust Storm
Warning Advisory and Assessment System (SDS-WAS), a project developed under the umbrella of the
World Meteorological Organization (WMO) with focus on improving capabilities of sand and dust storm
forecasts. For brevity reasons, only the main characteristics of the model are discussed here since a
thorough description is provided in Pérez et al. (2011, and references therein) as well as in recent
publications presenting its developments and applications in gas-phase chemistry (Badia et al., 2017),
volcanic ash dispersion (Marti et al., 2017) and data assimilation (Di Tomaso et al., 2017) studies. The
spectral variation of the GOCART dust optical properties, utilized as inputs to the radiation transfer
scheme, is presented in Section 3.2, whereas the model set up used in our experiments is given in Section

3.3.

*3.1. The NMMB-MONARCH model*

*3.1.1. The NMMB atmospheric model*

The Non-hydrostatic Multiscale Model NMMB (Janjic, 2004; Janjic and Black, 2007; Janjic et al.,
2011) is a unified atmospheric model developed at the National Centers for Environmental Prediction
(NCEP) (Janjic et al., 2001; Janjic, 2003). A powerful element of the model constitutes its non-
hydrostatic dynamical core, activated depending on the resolution, providing the capability to be used
for applications spanning at a wide range of temporal (from short- to long-term) and spatial (from
regional to global) scales. An additional dynamic feature of the NMMB is the consideration of various
parameterization schemes which can be incorporated into the numerical simulations. In our simulations,
the parameterization schemes of Betts-Miller-Janjic (Betts, 1986; Betts and Miller, 1986; Janjic, 1994,
2000), Ferrier (Ferrier et al., 2002), Mellor-Yamada-Janjic (Janjic et al., 2001) and Monin-Obukhov
(Monin and Obukhov, 1954) have been utilized for the convection, cloud microphysics, turbulence and





surface layer, respectively, as well as the NOAH land model (Ek et al., 2003). The model's dynamic
equations, in the horizontal plane, are solved on the Arakawa B grid (Arakawa and Lamb, 1977) while
in vertical the general hybrid pressure-sigma coordinate (Simmons and Burridge, 1981) is utilized. For
regional simulations, a rotated longitude-latitude coordinated system is used (the Equator is running
through the middle of the integration domain) enabling therefore more uniform grid distances.
*3.1.2.   The Dust component*

The main components of the desert dust life cycle, regarding mineral particles' production in the

source areas, transport and removal from atmosphere, are considered in the dust component of the
MONARCH model, which is embedded into the NMMB model. The size intervals as well as the effective
radii for each one of the 8 dust bins, representing clay-originated sub-micron (bins 1-4) and silt-originated
coarse (bins 5-8) particles, that are considered in the dust module were adopted from Pérez et al. (2006).
The mass of each bin is calculated at each time step, grid point and layer, while the median mass diameter
and the geometric standard deviation of the sub-bin distribution are fixed to 2.524 μm and 2.0 μm,
respectively. All the required parameters regulating dust emission and mobilization namely the: (i)
surface wind speed, (ii) turbulence, (iii) land use type, (iv) vegetation cover, (v) erodibility, (vi) surface
roughness, (vii) soil texture and (viii) soil moisture, are considered in the dust emission scheme (Pérez
et al., 2011). The vertical dust flux for each dust size bin is proportional to the horizontal sand flux while
several parameters are tuned to match observations that are mainly available far away from the sources.
Coarse dust aerosols are removed efficiently from the atmosphere through sedimentation, which is solved
implicitly in each model layer. For the description of dust aerosols' wet removal, a mechanism which is
more effective for fine mineral particles, parameterizations representing in- and below-cloud scavenging
are included in the NMMB-MONARCH in which the grid-scale cloud microphysical scheme of Ferrier
and the convective adjustment scheme of Betts-Miller-Janjic are utilized (Pérez et al., 2011). The ability
of the NMMB-MONARCH model to reproduce accurately the dust aerosol fields has been confirmed
through evaluation studies, relied on annual simulations, both at regional and global (Pérez et al., 2011)
scale as well as by utilizing measurements from experimental campaigns as reference data (Haustein et
al., 2012). Moreover, the reliability of the model in terms of reproducing the Saharan dust patterns over
Cape Verde as well as to simulate dust vertical profiles have been confirmed through the analyses made
by Gama et al. (2015) and Binietoglou et al. (2015), respectively. Finally, the predictive skills of the
NMMB-MONARCH model, in comparison with other regional models, have been assessed for a specific
dust outbreak (Huneeus et al., 2016) that affected the western parts of the Mediterranean and Europe.




### 3.2. Radiation transfer scheme and dust optical properties

For the description of dust aerosols interaction both with the shortwave (SW) and longwave (LW) radiation, the RRTMG (Rapid Radiative Transfer Model, Mlawer et al., 1997; Iacono et al., 2008) radiation transfer scheme is coupled with the dust module. RRTMG consists a modified version of the RRTM which is a broadband radiative transfer model that includes the molecular absorption of the SW (by water vapor, carbon dioxide, ozone, methane and oxygen) and LW (by water vapor, carbon dioxide, ozone, methane, nitrous oxide, oxygen, nitrogen and halocarbons) radiation. Even though the basic physics and absorption coefficients utilized in RRTM remain unchanged in RRTMG, several updates regarding computational efficiency and representation of subgrid-scale cloud variability have been considered (Iacono et al., 2008). Through these adjustments, it has been improved the efficiency of the RRTMG in global circulation model (GCM) applications with a minimal loss of accuracy (Iacono et al., 2008). In the RRTMG, the total number of quadratic points (g points) used to calculate radiances has been reduced from 224 to 112 and from 256 to 140 for the shortwave and longwave spectrum, respectively. In addition, for the short wavelengths, the discrete ordinates algorithm DISORT (Stammes et al., 1998) has been replaced by a two-stream radiation transfer solver (Oreopoulos and Baker, 1999). All the updates applied in the RRTMG radiation transfer code are listed in the Atmospheric and Environmental Research (AER) radiative transfer web site (http://www.rtweb.aer.com). Based on evaluation studies, the comparison of the RRTMG clear-sky SW and LW fluxes versus RRTM_SW and LBLRTM, respectively, has revealed that its accuracy at short wavelengths is within 3 $Wm^{-2}$ whereas at long wavelengths is 1.5 $Wm^{-2}$. As inputs to the radiation transfer scheme, the aerosol optical depth (AOD, measure of the aerosol load), the single scattering albedo (SSA, expresses the percentage of scattering to total extinction) and the asymmetry parameter (ASYM, expresses the angular distribution of the scattered radiation) are required. In the present version (v1.0) of the model, the calculation of dust optical properties is made based on the formulas presented in Pérez et al. (2006), by using the mass concentration simulated by the NMMB-MONARCH model and the single-particle optical properties derived by the GOCART model (Chin et al., 2002). The spectral variation of the single-particle optical properties for each bin, namely the mass extinction coefficient, the single scattering albedo and the asymmetry parameter are shown in Figures 2-i, 2-ii and 2-iii, respectively. Their calculation for each dust size bin and at each spectral band is made based on the Mie code (Mishchenko et al., 2002) assuming homogeneous and spherical dust particles. For the other types of tropospheric aerosols (sulfate, organic carbon, black carbon, and sea salt), the GOCART monthly climatological AOD, SSA and ASYM values





are utilized.
*3.3. Model set-up configuration*
In our experiments, the simulation domain (NMMB-MONARCH Simulation Domain, NSD, outer
domain in Figure 1) covers the Sahara (dust sources areas), the Mediterranean (mid-range dust transport
areas) as well as most of the European continent (long-range dust transport areas). The horizontal
resolution is equal to 0.25º x 0.25º degrees and 40 sigma-hybrid pressure levels up to 50 hPa are used in
vertical. The atmospheric model's fundamental time step is set to 25 seconds. The simulations have been
made for each one of the 20 identified Mediterranean desert dust outbreaks (see Section 2) considering
a spin-up and a forecast period, using 1º x 1º NCEP final analyses (FNL) as initial and 6-h boundary
conditions. More specifically, for each case, a hindcast period of 84 hours starts at 00 UTC of the day
(see the second column in Table 1) when the desert dust outbreak has been identified according to the
defined criteria (explained in Section 2). In order to ensure a more "realistic" initial state of the
atmosphere, a 10-day spin-up before the initialization of the forecast period is simulated, where the
model's meteorology is reinitialized every 24 hours. For the forecast periods, for the computation of the
dust radiative effects, two configurations of the model were run. In the first one (RADON), aerosols
interact with the short- and longwave radiation while in the second one the corresponding interactions
are deactivated (RADOFF).
**4.  Calculation of the dust direct radiative effects**
The direct radiative effects (DREs), expressed in $Wm^{-2}$, are computed at the top of the atmosphere
(TOA), into the atmosphere (ATM), and at the surface, for the downwelling (SURF) and the absorbed
(NETSURF) radiation, for the shortwave (SW), longwave (LW) and NET (SW+LW) radiation. The
calculations are made according to the following formulas:

$$DRE_{TOA} = F^{\uparrow}_{TOA,RADOFF} - F^{\uparrow}_{TOA,RADON} \ (Eq. \ 1)$$

$$DRE_{SURF} = F^{\downarrow}_{SURF,RADON} - F^{\downarrow}_{SURF,RADOFF} \ (Eq. \ 2)$$

$$DRE_{NETSURF} = \left(F^{\downarrow}_{SURF,RADON} - F^{\uparrow}_{SURF,RADON}\right) - \left(F^{\downarrow}_{SURF,RADOFF} - F^{\uparrow}_{SURF,RADOFF}\right) = F_{NETSURF,RADON} -$$
$$F_{NETSURF,RADOFF} \ (Eq. \ 3)$$






$DRE_{ATM} = DRE_{TOA} - DRE_{NETSURF}$ (Eq. 4)

At TOA (Eq.1), DREs are calculated through the subtraction of the RADON (dust-radiation
interaction is activated) from the RADOFF (dust-radiation interaction is deactivated) outputs of the
upward (↑) radiative fluxes ($F$) and express the loss (cooling effect or planetary cooling) or the gain
(warming effect or planetary warming) of energy within the Earth-Atmosphere system when are negative
and positive, respectively. At the surface, DREs are computed for both the downwelling (↓) (SURF, Eq.
2) and the net (downward minus upward) radiation (NETSURF, Eq. 3). Both DREs indicate a dust-
induced surface cooling or warming when they get negative or positive values, respectively. Finally, on
energy within the Earth-Atmosphere system, the $DRE_{ATM}$ is calculated by subtracting the $DRE_{NETSURF}$
from the $DRE_{TOA}$ values (Eq. 4) and quantifies the impact (warming or cooling) of dust outbreaks on the
atmospheric radiation budget. The DREs are based on the subtraction of two independent model runs.
Therefore, our results represent the radiative anomalies by dust aerosols including both the direct effect
and the fast response of atmospheric constituents such as humidity and clouds (semi-direct effects). DREs
are analyzed both at grid point (geographical distributions) and at regional scale levels and the obtained
results will be discussed in Section 5.2.
**5. Results**
*5.1. Comparison of model and satellite AODs*
Before dealing with the DREs, the ability of the model to reproduce satisfactorily the dust AOD fields
is assessed using MODIS-Terra $AOD_{550nm}$ retrievals as reference data. The results of the intercomparison
between the daily satellite AODs (left column in Fig. 3) and the modelled (right column in Fig. 3) dust
AODs at 12 UTC are presented here for three of the 20 identified desert dust outbreaks (see Section 2),
which took place on 2[nd] March 2005 (upper row in Fig. 3), 19[th] May 2008 (middle row in Fig. 3) and 2[nd]
August 2012 (bottom row in Fig. 3) and affected the eastern, central and western parts of the
Mediterranean basin, respectively. The corresponding maps for the remaining 17 cases are illustrated in
Figure S1. Note, that the evaluation of the model outputs versus the satellite measurements is restricted
within the geographical limits of the MSD (red rectangle in Fig. 1), since the satellite algorithm used for
identification of the desert dust outbreaks is applied only to this region (see Section 2).
According to the MODIS-Terra observations on 2[nd] March 2005, a dust plume extends from the Gulf
of Sidra to the southern parts of Greece, with AODs up to 5 (Fig. 3 i-a). As shown in Fig. 3 i-b, the model





is able to reproduce satisfactorily the spatial patterns of AOD on this day, with high dust $AOD_{550nm}$ values
(1-4) extending from Algeria to the Black Sea, affecting the eastern parts of the Mediterranean Sea. There
is a good agreement between the model and the satellite over areas where the satellite measurements are
available, highlighting the ability of the model to capture satisfactorily the spatial features of dust loads.
Nevertheless, several factors affect the level of agreement between model outputs and satellite
observations and for this reason only a qualitative intercomparison is attempted. The most important of
them are related to the differences regarding the spatiotemporal resolution and the aerosol optical depth
product. MODIS provides daily total AODs at 1° x 1° spatial resolution in contrast to the NMMB-
MONARCH model which produces instantaneous dust AODs at 0.25° x 0.25° spatial resolution.
Moreover, due to the inability of the MODIS Dark Target (DT) algorithm to retrieve aerosol optical
properties over desert areas as well as under cloudy conditions, in a significant part of the study region
there are not available satellite observations (white areas in Figs. 3 i-a, ii-a and iii-a) restricting thus their
comparison with the model outputs which provide full spatial coverage.
The second desert dust outbreak occurred on 19th May 2008 and affected the central sector of the
MSD. According to MODIS (Fig. 3 ii-a), the intensity of dust loads is maximized (up to 4) in the central
parts of the Mediterranean Sea (southeastern of Sicily). This is also reproduced by the model, although
somewhat higher AODs are found over the central and southern parts of Italy (Fig. 3 ii-b). In spite of
this, however, there is a clearly good model performance in reproducing the dust event that hit the central
Mediterranean. An ever better agreement between the model and satellite AODs, in terms of spatial
variability and intensity of dust loads, is found for the desert dust outbreak of August 2nd 2012, that
affected the westernmost parts of the Mediterranean, with highest AODs (up to 2-2.5) from the Alboran
Sea down to the coastal areas of Morocco (Figs. 3 iii-a,b). The model's ability to reproduce correctly the
spatial patterns and values of dust AODs is crucial for a successful computation of the dust DREs, since
DREs are determined to a large extent by AOD (e.g. Hatzianastassiou et al., 2004; Pérez et al., 2006;
Papadimas et al., 2012).

*5.2. Direct radiative effects (DREs)*

*5.2.1. Geographical distributions*

For each desert dust outbreak, the TOA, ATM, SURF and NETSURF DREs have been computed for
the SW, LW and NET radiation, according to the formulas presented in Section 4. Just as an example, in
Figure 4 are illustrated the geographical patterns of the instantaneous NET (SW+LW) $DRE_{TOA}$ (second
column), $DRE_{ATM}$ (third column), $DRE_{SURF}$ (fourth column) and $DRE_{NETSURF}$ (fifth column) values, at




12 h (first row), 24 h (second row), 36 h (third row) and 48 h (fourth row) after the initialization of the
model forecast on 2$^{nd}$ August 2012 at 00 UTC, along with the simulated patterns of dust AOD at 550 nm
on the same day and time (first column). For brevity reasons only the results for the all wave (NET) are
given, while the SW and LW DREs and their contribution to NET DREs are discussed in the regional
analysis (next sub-section). Moreover, for each desert dust outbreak, the minimum and maximum clear-
sky NET DREs at grid point level, during the simulation period, are presented in Table S1.

Based on the model outputs, at 12 h, an arc shaped dust plume affected the western parts of the Sahara,

the Canary Islands, the maritime areas off the Moroccan coasts, the southern parts of the Iberian
Peninsula and the western Mediterranean Sea (Fig. 4). During the forecast period, the spatial features of
the desert dust outbreak do not reveal a remarkable variability, with maximum AODs (up to 3) across
Mali, Mauritania, Western Sahara and in the Canary Islands. According to the simulated atmospheric
circulation patterns (results not shown here), strong near surface winds prevail across a convergence zone
(in western Sahara) developed between a low pressure system with its center in Mauritania-Mali and the
Azores subtropical anticyclone. At 700 hPa (~ 3000 meters), the uplifted mineral particles are transported
towards the western Mediterranean due to the prevailing strong southwesterly winds (~30 knots) off the
Moroccan coasts.  Similar synoptic conditions have been presented in Cluster 2 in Gkikas et al., (2015),
who studied the atmospheric circulation evolution related to the occurrence of desert dust episodes over
the Mediterranean.

At a first glance, it is evident that the DRE patterns are driven by those of the desert dust outbreaks

whereas small scale isolated features of extremely high/low DREs mainly result from slight "shifts" of
clouds between the two independent model runs. Moreover, it is apparent that both the sign and the
magnitude of DREs vary among TOA, surface and atmosphere as well as with time (day or night). It
must be mentioned that the limits of the DREs' colorbars in Figure 4 are set equal to -300 and 300 Wm$^-$
$^2$ in order to facilitate the intercomparison among the different levels within the Earth-Atmosphere
system, the comparison between day and night DREs and the visualization of our results. During daytime
(12 h and 36 h) the DREs are driven by their SW components which significantly exceed the LW ones.
Through absorption and scattering of solar radiation by mineral particles, the downwelling radiation at
the ground (SURF) is reduced by up to 308 Wm$^{-2}$, indicating a strong surface cooling (bluish colors) in
areas where the dust AOD is maximized like Mauritania or south Algeria. During nighttime (24 h and
48 h), the sign of the DRE$_{SURF}$ values is reversed and their magnitude decreases compared to that at 12
h and 36 h. This is because during the night the DRE$_{SURF}$ values are identical to the LW DRE ones, which
are positive, implying extra downwelling LW radiation at the surface, by up to 58 Wm$^{-2}$, emitted by the



overlying dust. This effect, leading to night surface warming, is more visible over specific parts of Sahara
that host high dust loads, e.g. in its western parts. The geographical patterns of $DRE_{NETSURF}$ are very
similar to those of $DRE_{SURF}$, as expected, since they only differ by the net upward radiation at the surface,
which in turn is determined by surface albedo and temperature. DRE for NETSURF expresses the amount
of radiation absorbed at the ground and is calculated through the subtraction of the upward from the
downward surface radiative fluxes (see Eq. 3). Therefore, the differences between $DRE_{SURF}$ and
$DRE_{NETSURF}$ are regulated by the upward component, which in turn is determined by the surface albedo
and temperature for the SW and LW radiation, respectively. For this reason, the negative differences (i.e.
$DRE_{SURF}$-$DRE_{NETSURF}$) at noon are maximized over highly reflective areas, while the positive ones at
night are observed in land areas where the surface cooling during sunlight hours is maximized (i.e.
reduction of the surface temperature during day leads to reduction of the emitted longwave radiation
during night). Based on our results, the negative (surface cooling) and positive (surface warming)
$DRE_{NETSURF}$ values can reach down to -290 $Wm^{-2}$ (eastern Atlantic Ocean) and up to 42 $Wm^{-2}$ (western
Sahara) during day and night, respectively. Among our studied cases (see Table S1) the instantaneous
NET $DRE_{SURF}$ and $DRE_{NETSURF}$ values at noon can be as large as -589 $Wm^{-2}$ and -337 $Wm^{-2}$, respectively,
in agreement with relevant results reported in previous studies dealing with the radiative impacts of dust
intrusions in the Mediterranean (Pérez et al, 2006; Remy et al., 2015), in west Africa (Heinold et al.,
2008; Mallet et al., 2009) and in Asia (Wang et al., 2009; Singh and Beegum et al., 2013).

The occurrence of desert dust outbreaks results in a strong perturbation of the atmospheric radiation

budget, attributed to the interaction of dust aerosols with the SW and LW radiation. More specifically,
during daytime (i.e. 12 h and 36 h), mineral particles absorb radiation at short wavelengths warming thus
the atmosphere as indicated by the positive instantaneous NET $DRE_{ATM}$ values in Figure 4 (third
column), reaching up to 189 $Wm^{-2}$ over the dust affected areas. Our calculated noon atmospheric DREs
(Table S1) are comparable to those reported by Heinold et al. (2008; 2011) and significantly lower
compared to those in Pérez et al. (2006), who found $DRE_{ATM}$ values higher than 500 $Wm^{-2}$ in land areas
with dust AOD > 3 during a desert outbreak that affected the Mediterranean on 12$^{th}$ April 2002. We note
that Pérez et al., (2006) used complex refractive indices taken from the Global Aerosol Data Set (GADS)
that have been shown to be excessively absorbing, which may partly explain their high $DRE_{ATM}$ values.
In order to highlight the strong instantaneous atmospheric warming induced by the desert dust outbreaks,
we have compared our results with similar ones obtained by previous studies that have been relied on
long-term model simulations. Zhao et al., (2011) found that the average net atmospheric warming across
the Sahara Desert, over the period April-September 2006, can be higher than 30 $Wm^{-2}$ based on regional





simulations of the WRF-Chem model. According to global simulations conducted at climatic scales (e.g.
Woodage and Woodward, 2014), dust aerosols can increase the absorbed radiation into the atmosphere
(warming effect) by up to 20 Wm$^{-2}$ across the Northern Africa. Radiative transfer computations of SW
DRE$_{ATM}$ for the 2000-2007 period by Papadimas et al. (2012) reported local values of a few decades up
to about 100 Wm$^{-2}$ in spring and summer above the Sahara Desert. During night, negative DRE$_{ATM}$ values
(down to -45 Wm$^{-2}$ in Algeria and Mali) are computed in the dust affected areas indicating an atmospheric
cooling because of the emission of LW radiation by mineral particles (Wang et al., 2013b).
The sign and magnitude of DRE$_{TOA}$ (Eq. 4) are regulated by DRE$_{NETSURF}$ and DRE$_{ATM}$. At noon and
above cloud-free areas, there is a distinct change of DRE$_{TOA}$ sign over oceanic and desert areas affected
by dust loads (note for example the red colors over the dusty western Sahara Desert regions, e.g.
Mauritania, against blue colors off the African coasts). This change of the DRE$_{TOA}$ sign is due to the
difference in surface albedo of the two types of surface (water and desert), in combination with dust high
AODs and low-to-moderate single scattering albedo enhancing solar absorption by dust above highly
multiple reflecting surfaces. Such a reverse of DRE$_{TOA}$ sign has been also reported in previous studies
(e.g. Santese et al., 2010; Nabat et al., 2012; Papadimas et al., 2012) and is characterized by a clear
contrast between red and blue colors (planetary warming and cooling, respectively) over adjacent
continental and oceanic areas. Over highly reflective surfaces (i.e. deserts), the atmospheric warming is
enhanced since dust aerosols absorb not only the incoming solar radiation but also the radiation reflected
by the surface. At the same time, the amount of the absorbed radiation at the ground is reduced by the
attenuation of the SW radiation and by the increase of the back reflected radiation at the surface. The
combination of these processes results in a predominance of the atmospheric warming over surface
cooling and subsequently to positive DRE$_{TOA}$ values (planetary warming), which can be as large as 85
Wm$^{-2}$ according to our simulations (Table S1). On the contrary, when dust aerosols are suspended over
dark surfaces (i.e. maritime areas), the condition is reversed and negative DRE$_{TOA}$ values down to -184
Wm$^{-2}$ (Table S1) are calculated, revealing thus a strong planetary cooling. Nevertheless, the positive
DRE$_{TOA}$ values exceeding 300 Wm$^{-2}$, which are recorded in maritime areas off the western African
coasts, are associated with the existence of absorbing dust aerosols superimposed over low- and mid-
level clouds. During night, the atmospheric cooling offsets the surface warming, both induced by the
desert dust outbreaks, and for this reason the DRE$_{TOA}$ values are almost negligible (do not exceed 10
Wm$^{-2}$ in absolute terms over cloud free areas) indicating an almost null dust direct radiative effect. Our
model computed dust induced planetary warming above western Africa is comparable to similar results
reported in previous studies focusing on the same or similar desert areas (e.g. Mallet et al., 2009; Pérez





et al., 2006; Wang et al., 2010; Nabat et al., 2012; Kalendeski and Stenchikov et al., 2016), although
differences also exist with regards to the magnitude or spatial $DRE_{TOA}$ patterns. These differences are
attributed to the different magnitude and spatial patterns of AOD values (dust loads) associated with the
different studied dust outbreaks, and also to differences in dust microphysical and optical properties
(Colarco et al., 2014).
*5.2.2.  Regional mean results*

In order to show more clearly temporal patterns, DREs were also averaged over the NSD (outer
NMMB Simulation Domain in Figure 1), SDD (Sahara Desert Domain, green rectangle in Figure 1) and
MSD (Mediterranean Satellite Domain, red rectangle in Figure 1) domains, for each desert dust outbreak,
separately for the NET, SW and LW radiation. Only cloud free grid points, in both model configurations
(RADON and RADOFF), with RADON dust $AOD_{550nm}$ values higher/equal than 0.05 have been
considered in this analysis. Then, in a further step, DRE values have been averaged over the 20 dust
outbreaks every three hours during the forecast period (84 hours). Thus, the time series of regional mean
and associated standard deviation (shaded areas) clear-sky TOA (black curve), SURF (purple curve),
NETSURF (blue curve) and ATM (red curve) DREs are depicted in Figure 5.
The SW clear-sky DREs (upper row in Fig. 5) are positive in the atmosphere (ATM, warming effect)
and negative at the surface (SURF and NETSURF, cooling effect) throughout the entire forecast period,
revealing a distinct diurnal cycle with marked maximum values around noon over all three domains. A
careful look, however, reveals some differences between the sub-regions. Thus, in NSD (first column)
and SDD (second column) the maximum $DRE_{ATM}$ values increase slightly with time from 30 to 35 $Wm^{-2}$,
while in contrast they decrease in MSD (third column) from 30 to 22 $Wm^{-2}$. Respectively, the negative
$DRE_{SURF}$ values (surface cooling) reach down to -50 $Wm^{-2}$ in the NMMB and the Sahara domain while
in the Mediterranean area reach down to -57 $Wm^{-2}$. In addition, the magnitude of $DRE_{SURF}$ and
$DRE_{NETSURF}$ values in NSD and SDD do not change with time, against a slight decrease in MSD to -40
$Wm^{-2}$. Hence, our results show that during the first 36 forecast hours, the computed $DRE_{SURF}$ and
$DRE_{NETSURF}$ values in MSD are larger than in SDD, while SDD is the source area of dust outbreaks. This
can be explained by the fact that the massive dust loads originating across the Sahara "enter" very fast
the Mediterranean domain leading thus to higher dust AODs (Figure S2) and larger reductions of surface
solar radiation ($DRE_{SURF}$ and $DRE_{NETSURF}$ values). Another possible explanation can be the variability
of dust loads' intensity across Sahara (SDD) and Mediterranean (MSD). In the former region, mineral
particles' loads are maximum but there are many grid points in which the simulated dust AODs are equal





or slightly higher than the required threshold (0.05). Therefore, the regional averages of DREs in SDD
are "smoothed" in contrast to MSD, where the number of grid points participating in the regional
calculations of DREs, although is lower, moderate-to-high dust AODs are forecasted there. As it concerns
the SW $DRE_{NETSURF}$ values, their temporal variation is identical to the corresponding ones for $DRE_{SURF}$;
however, the former ones are lower by up to 15 $Wm^{-2}$, in absolute terms. The most noticeable difference
between the two sub-domains (i.e. SDD and MSD) is encountered for the $DRE_{TOA}$ at noon. Over bright
desert surfaces, dust outbreaks warm the Earth-Atmosphere system as indicated by the positive $DRE_{TOA}$
values (up to 3.2 $Wm^{-2}$) while over the darker (mostly covered by sea) surfaces of the Mediterranean, the
mineral particles induce a planetary cooling with $DRE_{TOA}$ values ranging from -20 to -10 $Wm^{-2}$. In both
subdomains, the strongest planetary cooling is found at early morning and afternoon hours with negative
SW $DRE_{TOA}$ values down to -14.1 $Wm^{-2}$ and -25.1 $Wm^{-2}$ over Sahara and Mediterranean, respectively.
On the contrary, $DRE_{TOA}$ values decrease towards noon, due to increasing solar absorption and
decreasing scattering by dust under smaller solar zenith angles. Finally, due to the gradually decreasing
desert dust outbreaks' intensity within the MSD (Figure S2-i) for increasing forecast time, their radiative
impacts at all levels of the Earth-Atmosphere system are reduced as well in contrast to slightly increased
values in SDD.

The regional clear-sky DREs have been also computed for the LW spectrum (middle row in Figure 5)

revealing reverse effects of lower magnitude (in absolute terms) with respect to the corresponding ones
found at short wavelengths. Due to the emission of LW radiation by the mineral particles, desert dust
outbreaks induce an atmospheric cooling (negative LW $DRE_{ATM}$ values) and increase the amount of the
downward LW radiation at the surface (positive LW $DRE_{SURF}$ values). Both $DRE_{ATM}$ and $DRE_{SURF}$ levels
do not reveal remarkable temporal variation ranging from -4.8 to -2.8 $Wm^{-2}$ and from 1.9 to 4.3 $Wm^{-2}$,
respectively, in the NSD and SDD while slightly lower values are calculated for the MSD. On the
contrary, from the timeseries of the LW DREs for TOA and NETSURF it is evident the existence of a
diurnal cycle with maximum and minimum values around noon and during nighttime, respectively.
Moreover, both $DRE_{TOA}$ and $DRE_{NETSURF}$ values are higher than zero, throughout the simulation period,
indicating a warming LW radiative effect. More specifically, regional LW $DRE_{TOA}$ ranges from 0.1 to
3.4 $Wm^{-2}$ and $DRE_{NETSURF}$ varies between 2.9 and 8 $Wm^{-2}$ for the whole simulation domain (NSD). The
corresponding maximum DREs for the SDD and MSD are higher by up to 0.5 $Wm^{-2}$ and lower by up to
1.2 $Wm^{-2}$, respectively. Dust aerosols act like greenhouse gases (Miller and Tegen, 1998) trapping the
outgoing terrestrial radiation while at the same time emit radiation at longer wavelengths back to the
ground explaining thus the positive LW DREs for TOA (planetary warming) and NETSURF (surface



warming). In addition, the aforementioned LW DREs (TOA and NETSURF) covariate with time
revealing that the sign and the magnitude of the LW $DRE_{TOA}$ are determined by the perturbation of the
surface radiation budget (LW $DRE_{NETSURF}$) since the LW $DRE_{ATM}$ values are almost constant throughout
the simulation period. This is in contrast to the corresponding finding for the SW radiation where the
dust outbreaks' impact on the Earth-Atmosphere system's radiation budget is regulated by the
perturbation of the radiation fields into the atmosphere (ATM) and at the surface (NETSURF). Finally,
between SDD and MSD stronger LW DREs are found for the former domain due to the higher dust loads
over the Sahara as well as due to the larger size of mineral particles close to the source areas.

As it has been shown from the above analysis, the dust DREs between short and long wavelengths are

reverse (except at TOA over the Sahara around midday) and in order to assess the impact of desert dust
outbreaks in the whole spectrum the regional clear-sky NET (SW+LW) DREs have been also analyzed
(bottom row in Fig. 5). During sunlight hours, the NET DREs result from the compensation of the SW
and LW effects while during night the NET and the LW DREs are equal attributed to the absence of SW
radiation. Based on our results, in the NSD, the $DRE_{TOA}$, $DRE_{SURF}$, $DRE_{NETSURF}$ and $DRE_{ATM}$ range from
-13.9 to 2.6 $Wm^{-2}$, from -43.6 to 4 $Wm^{-2}$, from -26.3 to 3.9 $Wm^{-2}$ and from -3.7 to 28 $Wm^{-2}$, respectively.
In the SDD, the corresponding NET DREs vary from -11.9 to 7.1 $Wm^{-2}$, from -46.3 to 4.3 $Wm^{-2}$, from -
24.7 to 4.2 $Wm^{-2}$ and from -4.1 to 30.2 $Wm^{-2}$, respectively. Over the Mediterranean, the DREs for TOA
range from -23.7 to 0.9 $Wm^{-2}$, for SURF from -53.5 to 4.1 $Wm^{-2}$, for NETSURF from -39.4 to 4.2 $Wm^{-}$
$^{2}$ and for ATM from -4 to 25.5 $Wm^{-2}$. Moreover, due to the reduction of the desert dust outbreaks'
intensity for increasing forecast hours (Figure S2-i) the associated direct radiative effects are also reduced
within the MSD.

At noon, the SW planetary cooling counterbalances the LW planetary warming resulting thus to

almost zero $DRE_{TOA}$ values (null effect) over the whole simulation domain (NSD). On the contrary, in
the SDD, both SW and LW $DRE_{TOA}$ are positive due to the higher surface albedo and the trapping of the
surface upward LW radiation by mineral particles, respectively, leading to a net warming of the Earth-
Atmosphere system. In the broader Mediterranean area (MSD), the SW effects at TOA (planetary
cooling) dominate over the corresponding effects at longer wavelengths (planetary warming) explaining
thus the negative NET $DRE_{TOA}$ values (planetary cooling). In the atmosphere, for the three domains, the
negative LW DREs offset by about 12-14% the positive SW ones resulting to an overall warming effect
(positive NET $DRE_{ATM}$) around midday. Moreover, at noon, the increase of the absorbed LW radiation
at the ground offsets the decrease of the absorbed SW radiation by about 20-24% resulting in a NET
surface cooling (negative NET $DRE_{NETSURF}$) over the simulation domain. The corresponding levels for



the SDD and MSD vary from 25 to 28% and from 14 to 15%, respectively. In addition, the increase of
the downwelling LW radiation at the ground offsets by up to 8% the decrease of the downward SW
radiation resulting in negative NET DRE$_{SURF}$ values across the Sahara Desert (i.e. SDD).
Beyond the hourly and day-to-day variability of dust DREs, the results were averaged over the total
84-hour simulation period and the results are given, for the three domains, in Table 2, separately for the
SW, LW and NET radiation. The results are similar for the three domains, as to their physical meaning,
i.e. dust produces a SW cooling/heating of surface/atmosphere, resulting in a planetary SW cooling,
against a LW heating/cooling of surface/atmosphere, yielding a planetary LW heating. At TOA, desert
dust outbreaks cause a net planetary cooling with clear-sky NET DRE$_{TOA}$ values equal to -3.4±5.6, -
1.6±6.1 and -8.2±8.2 Wm$^{-2}$ for the NSD, SDD and MSD, respectively. Note, that due to the very strong
temporal variability of DREs at TOA, the computed standard deviations are considerably higher than the
averages in the NSD and SDD in contrast to MSD where are equal. Yoshioka et al., (2007), based on
long-term simulations, reported a negative DRE$_{TOA}$ (-4.73 Wm$^{-2}$) averaged over North Africa (mean dust
AOD equal to 0.39) and Heald et al. (2014) found that the all-sky NET DRE$_{TOA}$ values vary from -3 to
4 Wm$^{-2}$ across the Sahara for the year 2010. Woodage and Woodward (2014) and Zhao et al., (2011)
calculated positive DREs at TOA, averaged for the northwestern Africa, equal to 4.74 Wm$^{-2}$ and 0.83
Wm$^{-2}$, respectively. The negative averaged NET DRE$_{TOA}$ in SDD is attributed to the planetary cooling
found at early morning and afternoon hours. Wang et al. (2011) showed that when solar altitude is low
(i.e. high solar zenith angle) DRE at TOA is getting negative even over high-albedo deserts. Similar
results reported also by Banks et al. (2014), who studied the daytime cycle of dust DREs during the
Fennec campaign held in the central Sahara in June 2011. Our results for the DRE$_{TOA}$ in the MSD are
within the ranges reported in previous studies (e.g. Valenzuela et al., 2012; Sicard et al., 2014a;b) dealing
with dust intrusions in the Mediterranean. From the comparison of the SW and LW DRE$_{TOA}$, it is found
that the LW planetary warming offsets the SW planetary cooling by 33.3% in the NSD, by 52.9% in the
SDD and by 15.4% in the MSD. In the atmosphere, mineral particles cause an overall atmospheric
warming with NET DRE$_{ATM}$ levels varying from 7.3±11 (MSD) to 8.3±13 Wm$^{-2}$ (SDD) while the offset
of the SW atmospheric warming by the LW atmospheric cooling ranges from 31.1% to 33.6% among
the study domains. On an average, dust outbreaks reduce the downwelling NET radiation at the ground
(DRE$_{SURF}$) by up to -20.7±21.6 Wm$^{-2}$ (NSD), -20.1±21.9 Wm$^{-2}$ (SDD) and -22.7±23.6 Wm$^{-2}$ (MSD)
while the corresponding DRE$_{NETSURF}$ levels are equal to -11.6±13.4 Wm$^{-2}$, -9.9±12.4 Wm$^{-2}$ and -
15.6±17.5 Wm$^{-2}$, respectively. Our results for the SW and LW radiation in the SDD are in a good
agreement with the annual averages for the year 2008 presented by Nabat et al. (2012) over Northern



Africa. Santese et al. (2010), for a similar domain, calculated higher daily averages of regional SW and
LW $DRE_{NETSURF}$ values by up to 8 and 3 $Wm^{-2}$, respectively, for two dust outbreaks that took place on
17[th] and 23[rd] July 2003. For the $DRE_{SURF}$, the ratio of the LW/SW effects does not exceed 15.2% (SDD)
while the corresponding ratios for the $DRE_{NETSURF}$ are minimum for the MSD (23.5%) and maximum
for the SDD (37.7%). From the above analysis it is clear that dust outbreaks' impact at short wavelengths
is more pronounced than in the longwave spectrum; however, the contribution of the LW DREs to the
NET ones is significant or even larger, particularly over the Sahara at midday.

*5.3. Impact on sensible and latent heat fluxes*

As it has been shown in previous section, dust outbreaks exert a strong perturbation of the surface
radiation budget by reducing and increasing the absorbed NET radiation at the ground during day and
night, respectively. As a response to these disturbances, the surface heat fluxes, both sensible (SH) and
latent (LE), associated with the transfer of energy (heat) and moisture between surface and atmosphere,
also change in such a way trying to balance the gain or the loss of energy at the ground (Miller and Tegen,
1998). Subsequently, variations of SH and LE have impact on the components of the hydrological cycle
(Miller et al., 2004b) as well as on the turbulent kinetic energy and momentum transfer which in turn
affect near surface winds and dust emission (Pérez et al., 2006). Moreover, Marcella and Eltahir (2014)
and Kumar et al. (2014) have shown that due to the presence of dust aerosols into the atmosphere, the
daytime surface sensible heat fluxes are reduced leading to a reduction of the planetary boundary layer
(PBL) height.
Here, we are investigating the impact of desert dust outbreaks on SH and LE over the simulation
domain. It must be clarified that our analysis is restricted only above land areas since we are looking at
short-term effects and also in the existing version of the NMMB-MONARCH model the atmospheric
driver is not coupled with an ocean model. The time series of the regional SH and LE values, over the
forecast period, based on the RADON (red curve) and RADOFF (blue curve) configurations of the model
are presented in Figures 6-i (for SH) and 6-ii (for LE). Each curve corresponds to the mean levels
calculated from the 20 desert dust outbreaks while the shaded areas represent the associated standard
deviations. According to our results, SH is characterized by a diurnal variation with maximum values (~
350 $Wm^{-2}$) at noon and minimum ones (~ -30 $Wm^{-2}$) during nighttime (Fig. 6-i). Nevertheless, during
sunlight hours, the surface sensible heat fluxes simulated in the RADON experiment are lower by up to
45 $Wm^{-2}$ in comparison to the RADOFF outputs. At night, an opposite tendency is recorded and the



RADON SH fluxes are higher by up to 2 Wm$^{-2}$ than the corresponding fluxes based on the RADOFF
configuration of the model. The reverse effects on SH levels, over the western parts of the Sahara,
between daytime and nighttime as well as the diurnal variability of their magnitude have been pointed
out by Zhao et al. (2011). Based on the paired t-test, the differences between RADOFF and RADON SH
values are statistical significant at 95% confidence level throughout the forecast period. At local scale
(geographical distributions), among the studied cases, in areas where the desert dust outbreaks' intensity
is maximized, the SH fluxes are reduced by up to 150 Wm$^{-2}$ during day and increased by up to 50 Wm$^{-2}$
during night. Our findings are consistent with those presented by Mallet et al. (2009) and Rémy et al.
(2015) who analyzed the impact of dust storms on sensible heat fluxes over W. Africa and Mediterranean,
respectively, and substantially higher than the instantaneous perturbations of SH calculated by Kumar et
al. (2014), who studied a dust outbreak that occurred in northern India (17-22 April 2010).

The diurnal variation of the latent heat fluxes (Fig. 6-ii) is identical to that of sensible heat fluxes;

however, LE levels are remarkably lower than the regional averages of SH. This is attributed to the lower
soil water content and limited evaporation in arid regions leading thus to a predominance of the sensible
heat fluxes (Ling et al., 2014). Based on our simulations, LE values at noon gradually decrease both for
the RADOFF (blue) and RADON (red) experiments over the forecast period probably attributed to the
too moist initialization of the model (Note that the model is initialized with FNL analysis produced by a
different model (GFS)). Nevertheless, the latter LE values are lower than the former ones by up to 4 Wm$^{-2}$
indicating that desert dust outbreaks reduce the latent heat fluxes leaving from the ground. The
reliability of this finding is further supported by the fact that the RADOFF-RADON differences are
statistically significant at 95% confidence level. During night, the RADON LE values are slightly higher
(less than 0.5 Wm$^{-2}$) with respect to the corresponding ones simulated in the RADOFF configuration.
The instantaneous reduction and increase of LE (results not shown here) are substantially higher than the
regional perturbations and can be as large as -100 Wm$^{-2}$ and 20-30 Wm$^{-2}$, respectively. Finally, in contrast
to SH, the spatial features of LE anomalies are not identical with those of DRE$_{NETSURF}$ since other
parameters (e.g. soil moisture) regulate also the latent heat fluxes (Marcella and Eltahir, 2014).
*5.4. Impact on temperature fields*
Through the perturbation of the radiation by the mineral particles it is expected that desert dust

outbreaks will affect also the temperature fields. In order to quantify these impacts, the temperature
differences between the RADON and RADOFF simulations, both at 2 meters and in vertical, are
analyzed. In Figure 7, are displayed the RADON-RADOFF anomaly maps of temperature at 2 meters at





12 (i), 24 (ii), 36 (iii) and 48 (iv) hours after the initialization of the forecast period on 2$^{nd}$ August 2012
at 00 UTC. At noon, the highest negative biases (down to -4 K) are observed over land areas where the
intensity of dust loads is high (see the first and third row in the first column in Fig. 4) due to the strong
reduction of the NET radiation reaching the ground by the mineral particles. Similar findings, under dust
episode conditions, have been also reported by previous studies conducted for the Mediterranean (Pérez
et al., 2006), across the Sahara (Helmert et al., 2007; Heinold et al., 2008; Stanelle et al., 2010) and in
East Asia (Kumar et al., 2014; Ling et al., 2014). Over dust-affected maritime areas, due to the higher
heat capacity of the sea, the temperature differences between the RADON and RADOFF experiments
are almost negligible at these time scales. During nighttime, dust aerosols emit radiation at thermal
wavelengths increasing thus the near surface temperature when the dust-radiation interactions are
included into the numerical simulations (RADON experiment). For this reason, the RADON-RADOFF
temperature differences at 2 meters become positive (up to 4 K) at 24 and 48 forecast hours over land
areas where the "core" of the dust plume is observed. The reduction and the increase of the near surface
temperature during daytime and nighttime, respectively, either solely or as a combined result indicate
that the temperature diurnal range is reduced due to desert dust outbreaks.
The vertical distribution of dust layers determines their impacts on radiation with altitude which in
turn modify the temperature profiles (Meloni et al., 2015) and subsequently affect convection (Ji et al.,
2015), cloud development (Yin and Chen, 2007), precipitation (Yin et al., 2002) and wind profiles
(Choobari et al., 2012). In order to investigate the impacts of desert dust outbreaks on temperature fields
into the atmosphere, we have reproduced the altitude-latitude cross sections (up to 8 km above mean sea
level, m.s.l.) of RADON-RADOFF temperature differences on 4 April 2003 at 12 UTC along the
meridional 30° E (Fig. 8 ii-a) and on 7 March 2009 at 00 UTC along the meridional 10° E (Fig. 8 ii-b).
In addition, the corresponding cross sections of dust concentration (in kg m$^{-3}$) are shown in Figures 8 i-
a and 8 i-b, respectively. At midday, an elevated dust layer extends from 1.5 to 6 km m.s.l., between 23°
N and 33° N, with dust concentrations up to 0.8x10$^{-6}$ kg m$^{-3}$ while a low elevated dust layer extends from
the surface up to 1.5 km m.s.l., between 27° N and 31° N, with concentrations up to 10$^{-6}$ kg m$^{-3}$ (Fig 8 i-
a). Based on the cross section of temperature differences (Fig. 8 ii-a), dust aerosols via the absorption of
solar radiation warm the atmospheric layers by up to 0.8-0.9 K between altitudes where the high-elevated
dust layer is located. On the contrary, below the dust cloud, mineral particles cool the lowest tropospheric
levels (by up to 1.3 K) by attenuating (through scattering and absorption) the incoming solar radiation.
Note that between the parallels 31° N and 35° N, where dust loads are recorded at low altitudes (below
2 km), higher temperatures by up to 0.3 K are simulated in the RADON experiment with respect to



RADOFF, revealing thus an atmospheric warming near surface. Also, it must be considered that in this
area mineral particles are suspended over sea, where the impacts on sensible heat fluxes are negligible,
making therefore evident the dust warming effect at low atmospheric levels in contrast to land areas
(parallels between 27° N and 31° N), where the near surface temperature is reduced because of the
reduction of the sensible heat fluxes, as it has been shown also by Pérez et al. (2006, Fig. 10). Therefore,
the vertical distribution of dust loads plays a significant role regarding their impact on near surface
temperature which in turn may affect winds and subsequently dust emission (Stanele et al., 2010; Huang
et al., 2014). Above the high-elevated dust layer, negative RADON-RADOFF temperature differences
(down to -0.3 K) are found indicating an atmospheric cooling attributed to the dust albedo effect (Spyrou
et al., 2013).
In the second example, on 7[th] March 2009 at 00 UTC, a dust layer extends from the southern parts of
the NSD domain to the northern parts of Tunisia, between surface and 4 km m.s.l. (Fig. 8 i-b). Along the
dust plume, moderate concentrations (up to $0.5 \times 10^{-6}$ kg m$^{-3}$) are simulated between 15° N and 20° N,
low (less than $0.2 \times 10^{-6}$ kg m$^{-3}$) between 20° N and 25° N while the maximum ones (higher than $2 \times 10^{-6}$
kg m$^{-3}$) are recorded between 25° N and 35° N. Due to the emission of LW radiation by mineral particles,
dust aerosols cool the atmospheric layers (Otto et al., 2007) in which they reside, by up to 0.8 K, and
increase the temperature, by up to 0.4 K, just above the dust layer. Between the bottom of the dust layer
and surface, positive RADON-RADOFF temperature differences (i.e. warming) up to 1.2 K are
calculated as indicated by the red colors following the model topography (grey shaded). Mineral particles
emit LW radiation and trap the outgoing terrestrial radiation explaining thus the warming effect of dust
outbreaks close to the ground during nighttime. Nevertheless, this near surface warming is "interrupted",
being null or even reverse (i.e. cooling), in areas where the dust layer abuts the ground.

*5.5. Feedbacks on dust emission and dust aerosol optical depth*

In the present section, focus is given on the investigation of the potential feedbacks on dust AOD (at
550 nm) and dust emissions attributed to dust radiative effects. To this aim, the timeseries of the regional
averages and the associated standard deviations, throughout the forecast period (84 hours), calculated
from the 20 desert dust outbreaks for both parameters, based on the RADON (red) and RADOFF (blue)
experiments, are analyzed and the obtained results are shown in Figure 9. Over the simulation period,
the RADOFF dust AOD$_{550nm}$ gradually increases from 0.31 to 0.34 in contrast to the corresponding
outputs from RADON that are gradually decreasing down to 0.29 (Fig. 9-i). The positive RADOFF-
RADON differences of dust AOD, indicating a negative feedback when the dust-radiation interactions





are considered into the numerical simulations, are getting evident 12 hours (0.005 or 2%) after the
initialization of the forecast period and amplify with time (up to 0.036 or 12%), being also statistical
significant (paired t-test, confidence level at 95%) at each forecast step. The observed negative feedbacks
on dust AOD have been also pointed out in relevant studies carried out for specific desert dust outbreaks;
however, the reductions of mineral particles' loads were significantly higher compared to our
calculations corresponding to averages of the 20 studied desert dust outbreaks. More specifically, Pérez
et al. (2006) found a reduction of the regional dust AOD by up to 35-45 % and in Wang et al. (2010) the
corresponding reductions were ranging between 10 and 45%. Through the comparison of the mean dust
AOD levels, calculated over the 84-h simulation period, based on RADON (0.288) and RADOFF (0.308)
simulations, it is revealed a statistical significant reduction by 0.02 (6.9 %) attributed to the dust radiative
effects. Among the 20 desert dust outbreaks, these reductions vary from 1% (22 February 2004) to 12.5%
(27 January 2005) and are statistical significant at 95 % confidence level in all cases.
A similar analysis has been also made for the dust emissions (in kg m$^{-2}$) aggregated over the whole
simulation domain (NSD, outer domain in Figure 1). First, for each case and in each forecast step, the
dust emissions from all grid points within the NSD are aggregated. Then, the mean and the standard
deviation values are computed from the 20 desert dust outbreaks which are analyzed and the overall
results are given in Figure 9-ii. Moreover, the total dust emissions at each forecast step are added and the
obtained results, separately for the RADON and RADOFF configuration of the model, are provided into
the parentheses in the legend of Figure 9-ii. Dust emissions are maximized around midday (Cowie et al.,
2014) and are very weak during night. Based on the RADOFF simulation, the highest amounts of emitted
dust are increased from 2 to 2.5 kg m$^{-2}$ throughout the hindcast period. This increasing tendency is
encountered also in the RADON experiment; however, the dust emission is lower compared to the
simulation in which the dust-radiation interactions are neglected (RADOFF). The positive RADOFF-
RADON anomalies during daytime range from 0.1 to 0.4 kg m$^{-2}$ and are statistical significant at 95%
confidence level based on the paired t-test. Therefore, desert dust outbreaks exert a negative feedback on
dust emission explaining thus the reduction of dust AOD. The lower amounts of emitted dust, modelled
based on the RADON configuration, result from a chain of processes triggered by the surface cooling
which decreases the turbulent flux of sensible heat into the atmosphere, weakening the turbulent mixing
within the PBL and the downward transport of momentum to the surface and subsequently reduces
surface wind speed and dust emission (Miller et al., 2004a; Pérez et al., 2006).
During the simulation period, the total emitted amount of desert dust is equal to 18.279 and 21.849 kg
m$^{-2}$ based on the RADON and RADOFF, respectively. Therefore, desert dust outbreaks cause a negative





feedback on dust emissions reducing them by 3.57 kg m$^{-2}$ (-19.5%). This reduction (i.e. positive
RADOFF-RADON differences) is consistent in all the studied cases of our analysis varying from 0.6 kg
m$^{-2}$ (~10%, 24 February 2006) to 6.6 kg m$^{-2}$ (~34%, 2 August 2012). Negative feedbacks on dust AOD
and dust emissions have been also pointed out in previous studies based on short- (e.g. Ahn et al., 2007;
Rémy et al., 2015) and long-term (e.g. Perlwitz et al., 2001; Zhang et al., 2009) simulations. Woodage
and Woodward (2014) relied on climatic simulations of the HiGEM model, found a positive feedback
on global dust emissions which is in contradiction with findings reported in the majority of the existing
studies. The authors claimed that this discrepancy could be explained by the absence of mineral particles
with a radius larger than 10 μm in the emitted dust size distribution leading thus to an underestimation
of the LW effects. It must be clarified that according to our results negative feedbacks on dust emission
are found at a regional scale. Stanelle et al. (2010) showed that the vertical distribution of dust aerosols
determines their impacts on atmospheric stability and wind patterns and subsequently the associated
feedbacks on dust emissions which can be even positive at a local scale. This highlights the importance
of studying the potential feedbacks on mineral particles' loads as well as on their emissions spatially by
analyzing all the contributor factors.
*5.6. Assessment of the radiation at the ground*
The performance of the NMMB-MONARCH model in terms of reproducing the downward SW and
LW radiation is assessed using as reference data ground measurements derived from the Baseline Surface
Radiation Network (BSRN, Ohmura et al., 1998), a project of the Data and Assessments Panel from the
Global Energy and Water Cycle Experiment (GEWEX, http://www.gewex.org/) under the umbrella of
the World Climate Research Programme (WCRP, https://www.wcrp-climate.org/). Through this analysis
it is attempted to quantify objectively the potential improvements of the model's predictive skills
attributed to the inclusion of the dust radiative effects into the numerical simulations. Globally, 59 BSRN
stations are installed at different climatic zones providing radiation measurements (http://bsrn.awi.de/)
of high accuracy at very high temporal resolution (1 min) (Roesch et al., 2011). For the evaluation
analysis, we have used the global (direct and diffuse) shortwave and longwave downwelling radiation at
the ground measured at 6 stations (magenta star symbols in Figure 1) located in Spain (Izana, Cener),
France (Palaiseau, Carpentras), Algeria (Tamanrasset) and Israel (Sede Boker).
In Figure 10, are presented the timeseries of the measured (red curve) SW (i-) and LW (ii-) radiation
at Sede Boker and the corresponding model outputs based on the RADON (black curve) and the
RADOFF (blue curve) experiments, for the periods 22 February 2004 00 UTC – 25 February 2004 12



UTC (-a) and 21 April 2007 00 UTC – 24 April 2007 12 UTC (-b). In the bottom row of Fig. 10 are also
provided the temporal evolution of the model dust $AOD_{550nm}$ and the Level 2 AERONET total $AOD_{500nm}$
(red x symbols) retrieved via the O' Neill algorithm (O' Neill et al., 2003). Moreover, the AERONET
Ångström exponent (alpha) retrievals (denoted with green x symbols) are used as an indicator of coarse
or fine particles predominance into the atmosphere. For the comparison between model and observations,
the nearest grid point (minimum Euclidean distance) to the stations' coordinates is utilized. In Sede
Boker, the model's grid point elevation is 465 m being slightly lower than the AERONET (480 m) and
BSRN (500 m) stations, and therefore these small altitude differences do not affect substantially the
intercomparison results. Likewise, the SW and LW radiation are measured from 0.295 to 2.8 μm and
from 4 to 50 μm, respectively, while the spectral intervals in the model's radiation transfer scheme span
from 0.2 to 12.2 μm and from 3.3 to 1000 μm in the shortwave and longwave spectrum, respectively.
These differences might contribute to the level of agreement between model and observations; however,
are not discussed in our evaluation analysis.

In both examples presented here, but also for the rest of our dataset, the model captures better the

temporal variation of the downwelling SW in contrast to the LW radiation at the ground with correlation
coefficients (R) higher than 0.96 and between 0.63 and 0.85, respectively. However, the model-BSRN
biases vary strongly in temporal terms because of the inability of the model to reproduce adequately the
amount of the suspended mineral particles. For the first desert dust outbreak (left column in Fig. 10),
during the first forecast day, the maximum measured SW radiation is higher by about 150 $Wm^{-2}$ than the
simulated RADON outputs and slightly lower than the corresponding RADOFF levels. The former is
explained by the facts that the model reproduces the dust peak earlier than actually recorded according
to AERONET observations (see Figure 10 iii-a) and it develops low-level clouds (cloud fractions
between 0.5 and 0.6) while the latter one is attributed to the absence of dust radiative effects. For the rest
of the simulation period, the model overestimates and underestimates the shortwave and longwave
radiation, respectively, due to its deficiency to reproduce (underestimation) the amount of dust aerosols.
More specifically, based on AERONET retrievals, AOD and alpha levels vary from 0.2 to 0.4 and from
0.2 to 0.7, respectively, indicating the existence of dust loads of moderate intensity. On the contrary, the
simulated dust AOD at 550 is less than 0.1 in both model configurations characterized by a "flat"
behavior in temporal terms. Over the simulation period (22 February 2004 00 UTC – 25 February 2004
12 UTC), the mean SW (LW) radiation based on BSRN, RADON and RADOFF is equal to 221.6 $Wm^{-2}$
(290.0 $Wm^{-2}$), 255.4 $Wm^{-2}$ (266.4 $Wm^{-2}$) and 272.7 $Wm^{-2}$ (264.7 $Wm^{-2}$), respectively. Thanks to the
consideration of the dust radiative effects, the positive model-BSRN biases in the shortwave spectrum



are reduced from 51.1 Wm$^{-2}$ (RADOFF-BSRN) to 33.9 Wm$^{-2}$ (RADON-BSRN) while the negative
model-BSRN biases in the longwave spectrum are reduced from -25.3 Wm$^{-2}$ (RADOFF-BSRN) to -23.6
Wm$^{-2}$ (RADON-BSRN).

In the second case which is analyzed (right column in Fig. 10), two peaks are simulated with dust

AOD$_{550nm}$ values up to 0.9 (midday on 23$^{rd}$ April 2007) and 0.5 (afternoon on 21$^{st}$ April 2007). For the
major one, the model clearly overestimates aerosol optical depth with respect to AERONET retrievals in
which AOD (red x symbols) varies between 0.2 and 0.3 and alpha (green x symbols) ranges from 0.3 to
0.5 while the second one cannot be confirmed due to the lack of ground observations. Note, that between
09 UTC and 15 UTC on 23$^{rd}$ April 2007, the model underestimates the SW radiation by up to 200 Wm$^{-2}$
while overestimates the LW radiation by up to 150 Wm$^{-2}$ (maximum overestimations throughout the
simulation period) due to the misrepresentation of the dust AODs. Even higher model overestimations
of the SW radiation are observed at 12 UTC on 22 April 2007 attributed mainly to the inability of the
model to reproduce satisfactorily clouds, since the negative model-AERONET differences of AOD
cannot explain these large discrepancies in radiation. Clouds play an important role in such comparisons,
particularly when their features are not well reproduced by the model, leading to large overestimations
or underestimations, by up to 600 Wm$^{-2}$ in absolute terms among the studied cases of the present analysis,
as it has been pointed out in previous studies (e.g. Spyrou et al., 2013). Finally, the model (RADON)
overestimation of the SW radiation reaching the ground, by up to 200 Wm$^{-2}$ at 09 UTC on 21 April 2007,
is probably associated with underestimation of the simulated dust AOD since fair weather conditions are
forecasted and confirmed by the true color MODIS-Terra images (http://modis-
atmos.gsfc.nasa.gov/IMAGES/). For the SW radiation, the positive NMMB-BSRN biases during the
simulation period (21 April 2007 00 UTC – 24 April 2007 12 UTC) are reduced from 69.0 Wm$^{-2}$ to 40.9
Wm$^{-2}$ when dust-radiation interactions are activated (RADON) while lower positive biases for the LW
radiation are calculated (0.7 Wm$^{-2}$) when dust-radiation interactions are deactivated (RADOFF).
Summarizing, in the majority of the studied desert dust outbreaks here, positive and negative model-
observations biases are found for the downwelling SW (Table S1) and LW (Table S2) radiation,
respectively, which are reduced when the dust-radiation interactions are activated. On the contrary,
similar improvements are not evident on the correlation coefficients since are not found remarkable
differences between RADON-BSRN and RADOFF-BSRN R values (results not shown).







*5.7. Assessment of the temperature fields versus analysis datasets*

The forecasting performance of the NMMB-MONARCH model, except for the radiation (Section
5.6), has been also assessed for the temperature fields, utilizing as reference final analyses (FNL) derived
from the National Centers for Environmental Prediction database (http://rda.ucar.edu/). The evaluation
of both model configurations (RADON and RADOFF) against FNL temperature at 2 meters and at 17
pressure levels into the atmosphere is made at a regional scale for the NSD. For the former
intercomparison, only land grid points are taken into account since the atmospheric driver is not coupled
with an ocean model, while for the latter one it is not applied any criterion regarding the surface type
(land or sea). The evaluation of the model is made by considering grid points where the dust AOD is
higher/equal than 0.1, 0.5 and 1.0 representing low, moderate and intense dust load conditions,
respectively. In order to overcome spatial inconsistencies between model and analyses, the model outputs
have been regridded from their raw spatial resolution (0.25° x 0.25° degrees) to 1° x 1° degrees to match
FNL. We note that analyses datasets are only "best" estimates of the observed states of the atmosphere
and the surface produced by combining a model (in this case GFS) and available observations through
data assimilation techniques. Analysis datasets are more poorly constrained by observations over certain
regions including the arid and dusty ones, and more dependent on the model's behavior. This is even
more relevant for surface variables such as 2-m temperature which may heavily depend on the underlying
model's soil scheme.
In Figure 11, are presented the regional biases (model - FNL) of temperature at 2 meters for the
RADON (red curve) and RADOFF (blue curve) experiments, averaged from the 20 desert dust outbreaks
every 6 hours of the hindcast period, considering only land grid points where the dust AOD is
higher/equal than: (i) 0.1, (ii) 0.5 and (iii) 1.0. Regardless of the dust AOD threshold, cold biases are
found during night and early morning hours, warm biases are calculated in the afternoon while the
minimum biases in absolute terms appear at noon. According to our results, under low desert dust
conditions (Fig. 11-i), the agreement between model and FNL is better when the dust radiative effects
are neglected (RADOFF) during daytime, while slightly lower RADON-FNL biases compared to
RADOFF-FNL ones are found during night. At noon, the RADOFF-FNL biases are almost zero (less
than 0.l K) whereas negative RADON-FNL biases (down to -0.27 K) are computed due to the surface
cooling induced by the mineral particles. For moderate dust AODs (Fig. 11-ii), during night, the model-
FNL temperature biases are lower for the RADON configuration (less than 1 K) in contrast to the
RADOFF simulation (less than 1.4 K) and these improvements are statistically significant at 95%





confidence level. Nevertheless, at midday, the RADOFF-FNL biases are similar to those found for the
lowest dust AOD threshold (Fig. 11-i), while the model cold biases, varying from -1.15 K (84 h) to -0.55
K (12 h), are amplified when the dust-radiation interactions are activated (RADON). The "corrections"
of the near surface temperature forecasts during nighttime become more evident and statistically
significant, when only land areas affected by intense dust loads (dust AOD ≥ 1.0) are considered in the
NMMB-FNL comparison. Under these high dust AODs, the increase of air temperature at 2 meters due
to the dust LW DREs reduces the existing cold biases. Therefore, the improvements on model's
predictability of temperature at 2 meters when accounting for dust-radiation interactions, are more
evident when the intensity of dust loads increases.
The potential impacts of the dust radiative effects inclusion on the model's forecasting ability have
been also investigated for the temperature fields in vertical. For this purpose, from the 20 desert dust
outbreaks, the temperature model-FNL biases at 17 pressure levels (from 1000 to 100 hPa) have been
calculated in RADOFF (black curve) and RADON (red curve) and the obtained results are illustrated in
Figure 12. The assessment results are presented only 24 (a) and 48 (b) hours after the initialization of the
forecast period since are not found remarkable differences between the two model configurations at noon
(i.e. 12 and 36 UTC).
Based on our findings, model warm biases are found between 950 and 700 hPa where most of the dust
is confined (brown curve). For the lowest dust AOD threshold, these positive model-FNL biases reach
up to 0.245 K and 0.313 K at 24 and 48 forecast hours, respectively, when mineral particles are not
treated as radiatively active substance (RADOFF). On the contrary, when dust-radiation interactions are
activated (RADON) the corresponding biases are reduced down to 0.155 K and 0.239 K, respectively,
indicating a better model performance which is further supported by the fact that these improvements are
statistical significant (95 % confidence level). Similar but more evident results are found when the dust
AOD threshold increases from 0.1 to 0.5 (middle row in Figure 12). More specifically, at 24 forecast
hours, the RADON-FNL temperature differences do not exceed 0.321 K in contrast to the corresponding
biases between RADOFF and FNL which can be as high as 0.512 K. At 48 forecast hours, between
altitudes where the dust concentrations are maximized, the red curve (RADON-FNL) is close to the blue
thick line which represents the ideal score (i.e. zero biases), while the RADOFF warm biases can reach
up to 0.443 K. As it has been shown in Section 5.4 (see Fig. 8 ii-b), due to the emission of longwave
radiation by the mineral particles there is a temperature reduction within the atmospheric layers in which
they are confined and a slight warming above the dust layer. The former effect explains the statistically
significant reduction of the model warm biases between 950 and 700 hPa whereas the latter one could



explain the slight statistically significant reduction of the model cold biases recorded between 600 and
500 hPa (see Fig. 12 ii-a). For the highest dust AOD threshold, at 24 forecast hours (Fig. 12 iii-a), the
agreement of temperature profiles between RADON and FNL is better compared to RADOFF-FNL
whereas at 48 forecast hours depends on altitude (Fig. 12 iii-b). Summarizing, thanks to the consideration
of the dust radiative effects the predictive skills of the NMMB-MONARCH model in terms of
reproducing temperature fields within the atmosphere are improved. Our findings are in agreement with
previous studies in which similar evaluations have been made either against analyses datasets (Pérez et
al., 2006) or weather station observations (Wang et al., 2010; Wang and Niu, 2013).
**6.   Summary and conclusions**

In the present study, the direct radiative effects (DREs) induced by 20 intense and widespread
Mediterranean desert dust outbreaks, that took place during the period March 2000 – February 2013,
have been analyzed based on short-term (84 hours) regional simulations of the NMMB-MONARCH
model. The identification of desert dust outbreaks has been accomplished via an objective and dynamic
algorithm (Gkikas et al., 2013; 2016) utilizing as inputs daily satellite retrievals, available at 1° x 1°
spatial resolution, providing information about aerosols' load (*AOD*), size (*FF, α*) and nature (*AI*). DREs
have been calculated at the top of the atmosphere (TOA), into the atmosphere (ATM), and at the surface,
for the downwelling (SURF) and the absorbed (NETSURF) radiation, for the shortwave (SW), longwave
(LW) and NET (SW+LW) radiation. The obtained results have been presented through geographical
distributions as well as at regional level by averaging the clear-sky DREs over the whole simulation
domain (NSD), the Sahara Desert (SDD) and the broader Mediterranean basin (MSD) sub-domains. At
a further step, the impacts of desert dust outbreaks on sensible and latent heat fluxes as well as on
temperature at 2 meters and into the atmosphere have been investigated. Moreover, the potential
feedbacks on dust emission and dust AOD, attributed to dust-radiation interactions, have been assessed
at regional scale representative for the simulation domain used in our experiments. In the last part of our
study, focus was given on the potential improvements on model's predictive skills, attributed to the
inclusion of dust radiative effects into the numerical simulations, in terms of reproducing the downward
SW/LW radiation at the ground as well as the temperature fields. The main findings obtained from the
present analysis are summarized below.





**Direct Radiative Effects**

➢ DREs into the atmosphere and at the surface are driven by the dust outbreaks' spatial features whereas at TOA, the surface albedo plays a crucial role, particularly under clear sky conditions.

➢ At noon, dust outbreaks induce a strong surface cooling with instantaneous NET $DRE_{SURF}$ values down to -589 $Wm^{-2}$.

➢ Similar spatial patterns are revealed for the absorbed radiation at the ground; however, the magnitude of the NET $DRE_{NETSURF}$ values is lower in comparison with the corresponding DREs for SURF.

➢ Through the absorption of the incoming solar radiation by the mineral particles, dust outbreaks can increase the atmospheric radiation budget (warming effect) by up to 319 $Wm^{-2}$ around midday.

➢ At TOA, positive DREs up to 85 $Wm^{-2}$ are found over highly reflective surfaces indicating a planetary warming while negative DREs down to -184 $Wm^{-2}$ are computed over dark surfaces indicating a strong planetary cooling.

➢ During nighttime, reverse effects of lower magnitude are found into the atmosphere and at the surface with maximum instantaneous NET $DRE_{SURF}$, $DRE_{NETSURF}$ and $DRE_{ATM}$ values equal to 83 $Wm^{-2}$, 50 $Wm^{-2}$ and -61 $Wm^{-2}$ whereas at TOA due to the offset of the atmospheric cooling by the surface warming, the $DRE_{TOA}$ values are almost negligible (less than 10 $Wm^{-2}$).

➢ The regional NET clear-sky DREs for the NSD range from -13.9 to 2.6 $Wm^{-2}$, from -43.6 to 4 $Wm^{-2}$, from -26.3 to 3.9 $Wm^{-2}$ and from -3.7 to 28 $Wm^{-2}$ for TOA, SURF, NETSURF and ATM, respectively.

➢ For the regional clear-sky NET DREs at TOA, the calculated positive DREs (7.1 $Wm^{-2}$) in the SDD and the negative DREs (-15 $Wm^{-2}$) in the MSD at noon indicate a planetary warming and cooling in the Sahara and in the Mediterranean, respectively.

➢ Over the 84 hours forecast period, the LW surface warming offsets by up to 37.7% the SW surface cooling whereas the LW atmospheric cooling offsets by 33.6% the SW atmospheric warming.

➢ At TOA, the corresponding LW/SW ratios vary from 15.4% (MSD) to 52.9% (SDD); however, the contribution of the LW DREs to the NET ones is comparable or even larger, particularly over the Sahara at midday.





**Sensible and latent heat fluxes**

> As a response to the surface radiation budget perturbations, desert dust outbreaks reduce the sensible heat fluxes (regional averages taking into account only land grid points) by up to 45 Wm$^{-2}$ during daytime while reverse tendencies of lower magnitudes are found during night (2 Wm$^{-2}$).

> Locally, the aforementioned values can reach down to -150 Wm$^{-2}$ and up to 50 Wm$^{-2}$.

> At noon, dust outbreaks reduce also the surface latent heat fluxes by up to 4 Wm$^{-2}$ and 100 Wm$^{-2}$ at a regional and grid point level, respectively. At night, the regional and the instantaneous LE levels are increased by up to 0.5 Wm$^{-2}$ and 30 Wm$^{-2}$, respectively.

**Impact on temperature fields**

> Due to the attenuation of the incoming solar radiation and the emission of radiation at thermal wavelengths, both induced by dust aerosols, temperature at 2 meters reduces and increases during day and night, respectively, by up to 4 K in absolute terms in land areas where the dust loads are maximized (AODs higher than 2).

> At noon, dust outbreaks warm the atmosphere by up to 0.9 K between altitudes where elevated dust layers are located and cool the lowest tropospheric levels by up to 1.3 K, due to the reduced surface sensible heat fluxes.

> Due to the emission of LW radiation and the trapping of the outgoing terrestrial radiation by dust aerosols, the nocturnal temperature decreases by up to 0.8 K in atmospheric altitudes where mineral particles are confined, whereas between the bottom of the dust layer and the surface, the air-temperature increases by up to 1.2 K.

**Feedbacks on dust AOD and dust emission**

> The total emitted amount of dust is reduced by 19.5% (statistically significant at 95% confidence level) over the forecast period when dust DREs are included into the numerical simulations, revealing thus a negative feedback on dust emissions.

> Among the studied cases, the corresponding percentages range from -34% (2 August 2012) to -10% (24 February 2006) and are statistical significant (95% confidence level) in all cases.





- ➢ As a consequence of the lower amount of mineral particles emitted in the atmosphere, negative feedbacks are also found on the mean regional dust $AOD_{550nm}$ which is decreased by 0.02 (6.9%) with respect to the control experiment (RADOFF).
- ➢ Statistically significant reductions of the regional dust $AOD_{550nm}$, varying from 1% (22 February 2004) to 12.5% (27 January 2005), are found in all the studied cases when dust-radiation interactions are activated (RADON).

**Assessment of model's predictive skills**

- ➢ Through the evaluation of the model's forecast outputs of the downwelling SW and LW radiation at the ground against surface measurements derived by the BSRN network, it is revealed a reduction of the modelled positive and negative biases for the SW and LW radiation, respectively, attributed to the consideration of the dust radiative effects. However, model's accuracy is critically affected by its ability to represent satisfactorily clouds' spatiotemporal features highlighting thus the key role of other model errors when such comparisons are attempted.
- ➢ Under high dust load conditions (AODs higher/equal than 0.5), the nocturnal model-FNL negative regional biases of temperature at 2 meters are reduced by up to 0.5 K (95% statistically significant) in the RADON experiment. On the contrary, these temperature "corrections" are not evident during daytime revealing thus that other model errors (particularly those introduced by the soil model) can dominate over the expected improvements attributed to the consideration of dust-radiation interactions in the numerical simulations.
- ➢ The model regional warm biases found at 24 and 48 hours after the initialization of the forecast period, between pressure levels (950 and 700 hPa) where the dust concentration is maximized, are reduced by up to 0.4 K (95% statistically significant) in the RADON experiment.

**Acknowledgments**

The MDRAF project has received funding from the European Union's Seventh Framework Programme for research, technological development and demonstration under grant agreement no 622662. O. Jorba and S. Basart acknowledge the grant CGL2013-46736 and the AXA Research Fund. C. Pérez García-Pando acknowledges long-term support from the AXA Research Fund, as well as the support received through the Ramón y Cajal programme (grant RYC-2015-18690) of the Spanish Ministry of Economy and Competitiveness. Simulations were performed with the Marenostrum Supercomputer at the



Barcelona Supercomputing Center (BSC). We would like to thank the principal investigators maintaining
the BSRN sites used in the present work. The authors would like thank the Arnon Karnieli for his effort
in establishing and maintaining SEDE_BOKER AERONET site.

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



**Table 1**: List of the Mediterranean desert dust outbreaks which have been identified based on the satellite algorithm. In
addition, the number of DD episodes (number of satellite grid cells at 1° x 1° spatial resolution where a DD episode has been
identified), the regional intensity (in terms of $AOD_{550nm}$) calculated from the DD episodes as well as the dust affected parts of
the Mediterranean domain are provided.

| Case | Date | DD episodes | Intensity | Affected parts of the Mediterranean domain |
|---|---|---|---|---|
| 1 | 31 July 2001 | 85 | 0.74 | Western |
| 2 | 8 May 2002 | 71 | 1.60 | Central |
| 3 | 4 April 2003 | 53 | 1.42 | Eastern |
| 4 | 16 July 2003 | 83 | 0.98 | Western and Central |
| 5 | 22 February 2004 | 46 | 2.18 | Central and Eastern |
| 6 | 26 March 2004 | 66 | 1.45 | Central and Eastern |
| 7 | 27 January 2005 | 37 | 1.36 | Central and Eastern |
| 8 | 2 March 2005 | 45 | 2.96 | Central and Eastern |
| 9 | 28 July 2005 | 30 | 1.08 | Western and Central |
| 10 | 24 February 2006 | 45 | 2.92 | Eastern |
| 11 | 19 March 2006 | 39 | 1.37 | Eastern |
| 12 | 24 February 2007 | 42 | 2.29 | Central and Eastern |
| 13 | 21 April 2007 | 42 | 1.65 | Central |
| 14 | 29 May 2007 | 47 | 1.40 | Eastern |
| 15 | 10 April 2008 | 42 | 1.58 | Central |
| 16 | 19 May 2008 | 66 | 1.45 | Central |
| 17 | 23 January 2009 | 36 | 2.65 | Eastern |
| 18 | 6 March 2009 | 41 | 1.41 | Eastern |
| 19 | 27 March 2010 | 39 | 1.43 | Central |
| 20 | 2 August 2012 | 35 | 1.20 | Western |






**Table 2:** Mean and standard deviation of clear-sky $DRE_{TOA}$, $DRE_{SURF}$, $DRE_{NETSURF}$ and $DRE_{ATM}$ values, over the simulation
period (84 hours), calculated in the NSD, SDD and MSD domains for the SW, LW and NET radiation. Blue and red
background colors indicate negative (cooling effect) and positive (warming effect) DREs, respectively.

| | | $DRE_{TOA}$ | $DRE_{SURF}$ | $DRE_{NETSURF}$ | $DRE_{ATM}$ |
|---|---|---|---|---|---|
| NSD | SW | -5.1±5.9 | -24±21.1 | -17.1±15 | 12±12.6 |
| | LW | 1.7±1.3 | 3.3±0.7 | 5.5±1.8 | -3.8±0.7 |
| | NET | -3.4±5.6 | -20.7±21.6 | -11.6±13.4 | 8.2±12.2 |
| SDD | SW | -3.4±6.1 | -23.7±21.3 | -15.9±14.3 | 12.5±13.4 |
| | LW | 1.8±1.5 | 3.6±0.8 | 6±2.2 | -4.2±0.8 |
| | NET | -1.6±6.1 | -20.1±21.9 | -9.9±12.4 | 8.3±13 |
| MSD | SW | -9.7±8.8 | -25.8±23.2 | -20.4±18.5 | 10.6±11.2 |
| | LW | 1.5±0.8 | 3.1±0.8 | 4.8±1.2 | -3.3±0.6 |
| | NET | -8.2±8.2 | -22.7±23.6 | -15.6±17.5 | 7.3±11 |









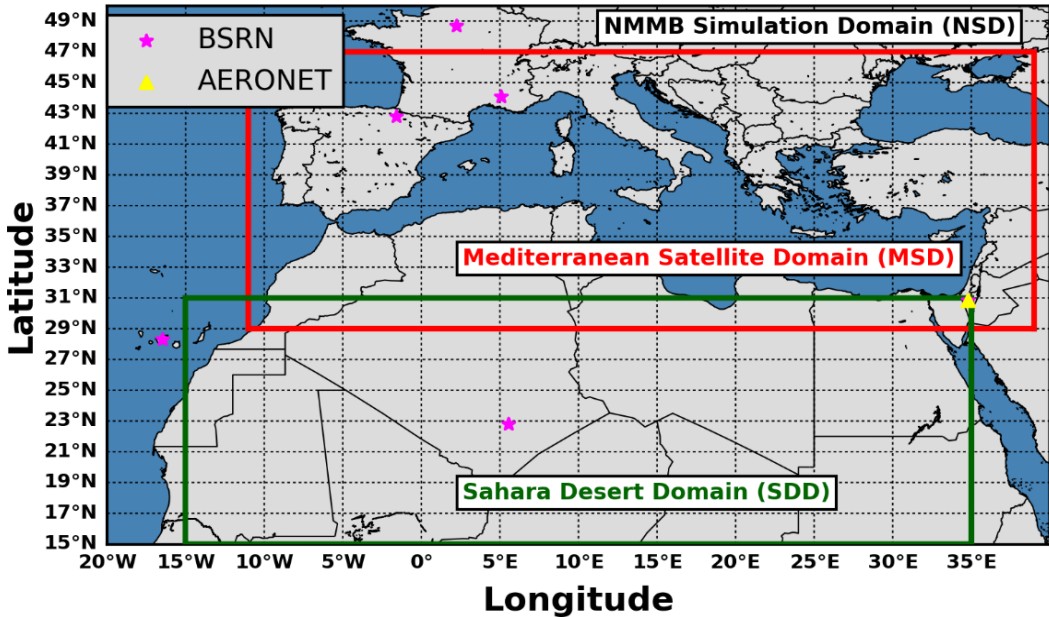

**Figure 1:** Geographical limits of the: (i) NMMB Simulation Domain (*NSD*, outer domain), (ii) Mediterranean Satellite
Domain (*MSD*, red rectangle) and (iii) Sahara Desert Domain (*SDD*, green rectangle). With the magenta star symbols are
depicted the locations of the BSRN stations and with the yellow triangle is denoted the location of the AERONET Sede Boker
station.












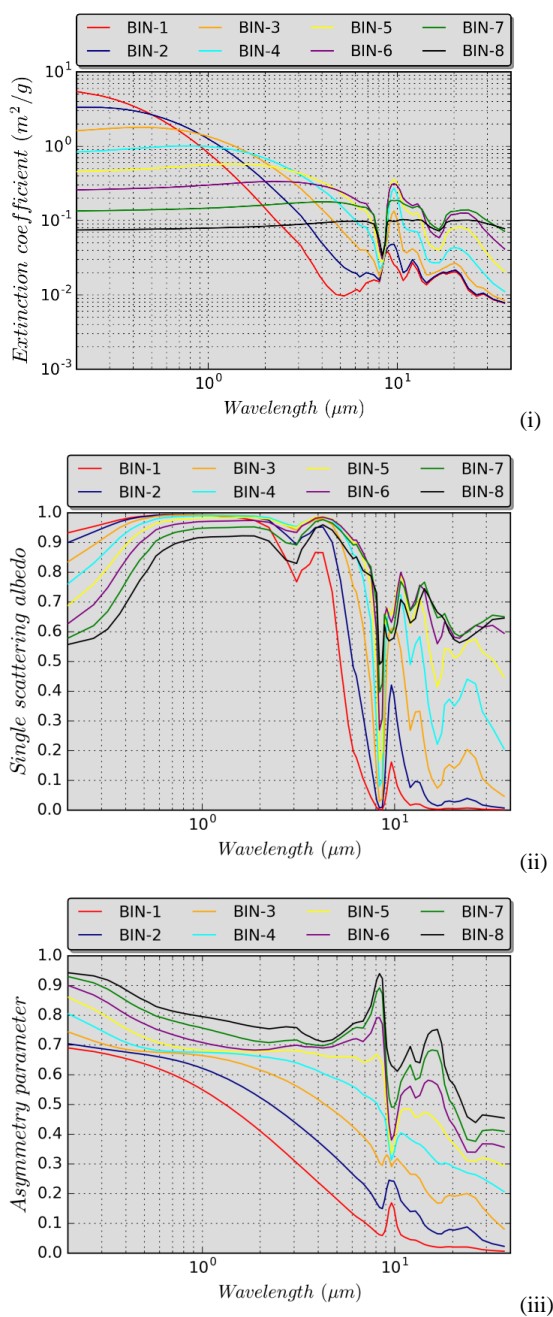

**Figure 2:** Spectral variation of the GOCART: (i) extinction coefficient (in m²/g), (ii) single scattering albedo and (iii) asymmetry parameter, for each one of the 8 dust bins which are considered in the dust module.



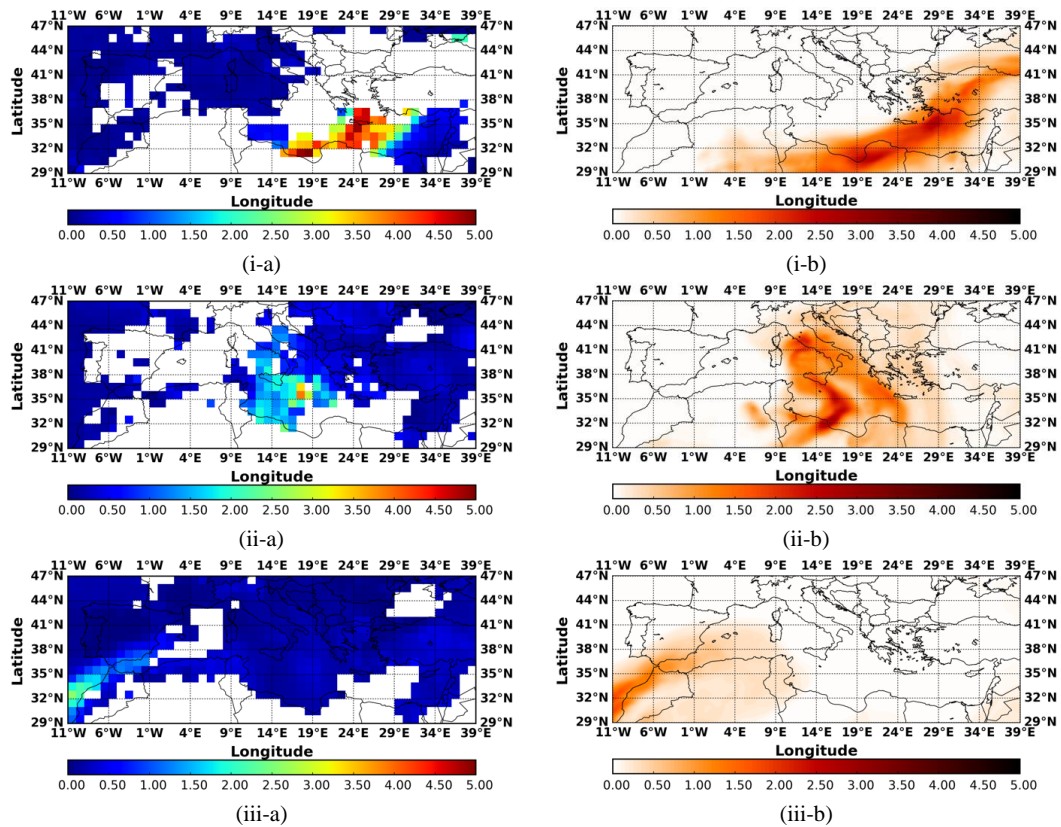

**Figure 3:** Geographical distributions of: (a) the daily averaged MODIS-Terra AOD at 550nm and (b) the simulated dust AOD at 550nm at 12:00 UTC for the Mediterranean desert dust outbreaks that took place on: (i) 2nd March 2005, (ii) 19th May 2008 and (iii) 2nd August 2012.



**Figure 4:** Spatial patterns of the simulated dust AOD$_{550nm}$ and the instantaneous DRE$_{TOA}$, DRE$_{ATM}$, DRE$_{SURF}$ and DRE$_{NETSURF}$ values, expressed in Wm$^{-2}$, at 12, 24, 36 and 48 hours after the initialization of NMMB-MONARCH model at 00 UTC on 2$^{nd}$ August 2012.



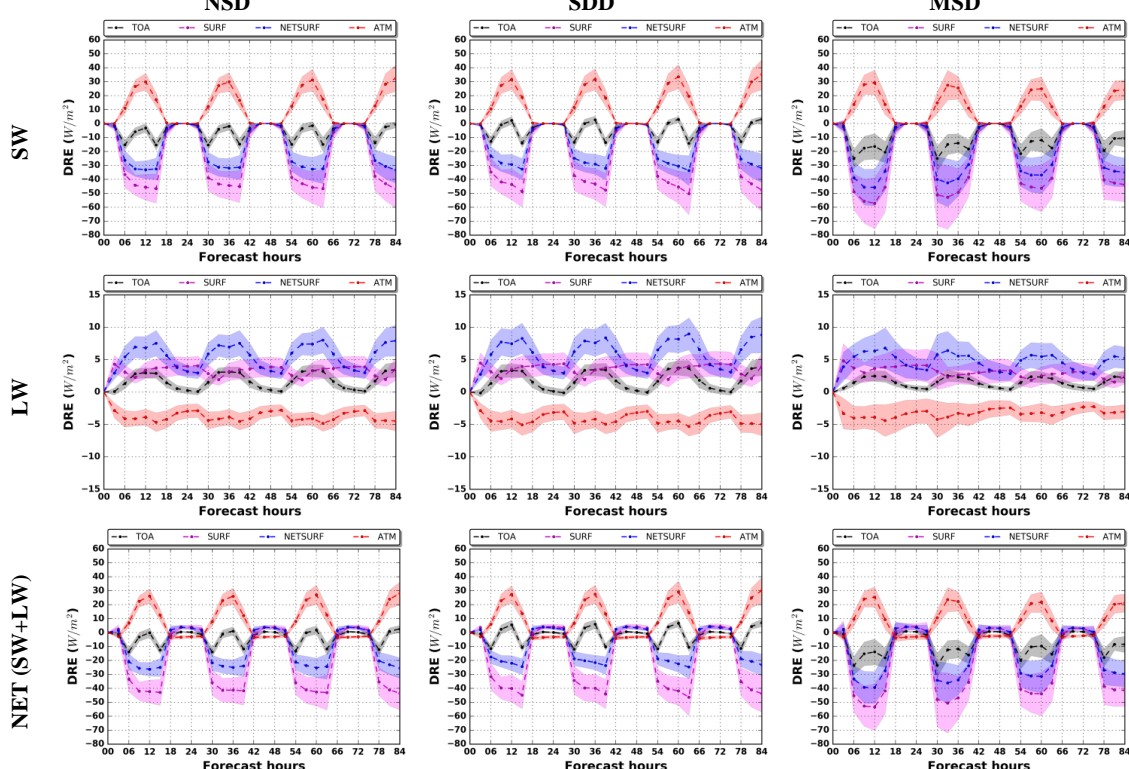

**Figure 5:** Regional clear-sky SW (upper row), LW (middle row) and NET (SW+LW) (bottom row) DREs at TOA (black), SURF (purple), NETSURF (blue) and ATM (red) averaged over the NSD (left column), SDD (central column) and MSD (right column) domains. The calculated DREs correspond to the mean values calculated by the 20 simulated Mediterranean desert dust outbreaks and the shaded areas represent the associated standard deviation.





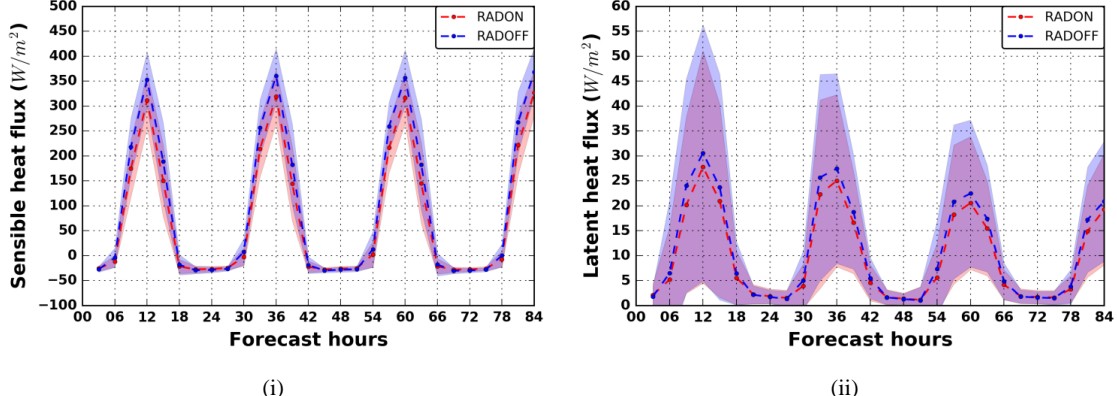

|     |     |
|:---:|:---:|
| (i) | (ii) |

**Figure 6:** Regional averaged values, over land areas of the simulation domain affected by dust loads and under clear-sky conditions, of the: (i) sensible and (ii) latent heat fluxes, expressed in Wm$^{-2}$, based on the RADON (red) and the RADOFF (blue) configuration of the NMMB-MONARCH model. The dashed lines correspond to the mean values calculated by the 20 simulated Mediterranean desert dust outbreaks and the shaded areas represent the associated standard deviation.





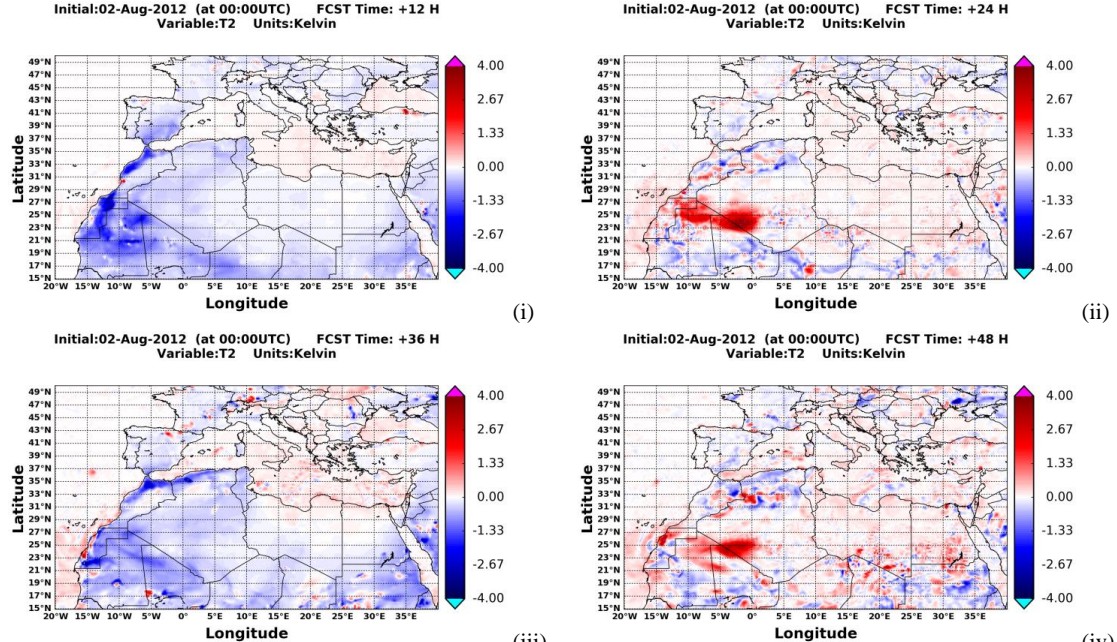

**Figure 7:** Spatial patterns of temperature differences at 2 meters, between the RADON and RADOFF configuration of the NMMB-MONARCH model, for the: (i) 12, (ii) 24, (iii) 36 and (iv) 48 hours forecast of the 00 UTC cycle on 2nd August 2012.





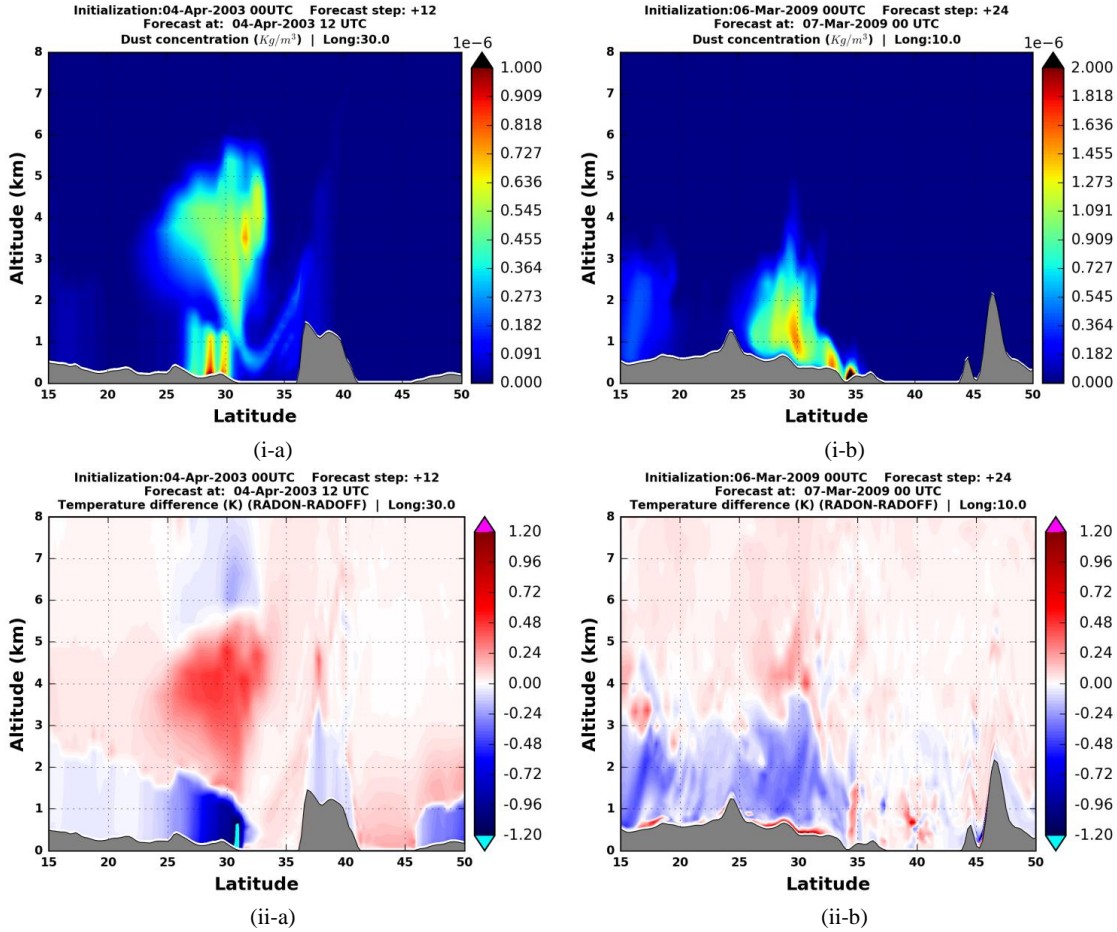

(i-a)                    (i-b)

(ii-a)                   (ii-b)

**Figure 8:** Altitude-latitude cross-sections (up to 8 km m.s.l.) simulated by the NMMB-MONARCH model of the: (i) dust concentration (in kg m$^{-3}$) and (ii) RADON-RADOFF temperature anomalies (in K) on: (a) 4 April 2003 at 12 UTC along the meridional 30° E and (b) 7 March 2009 00 UTC along the meridional 10° E.





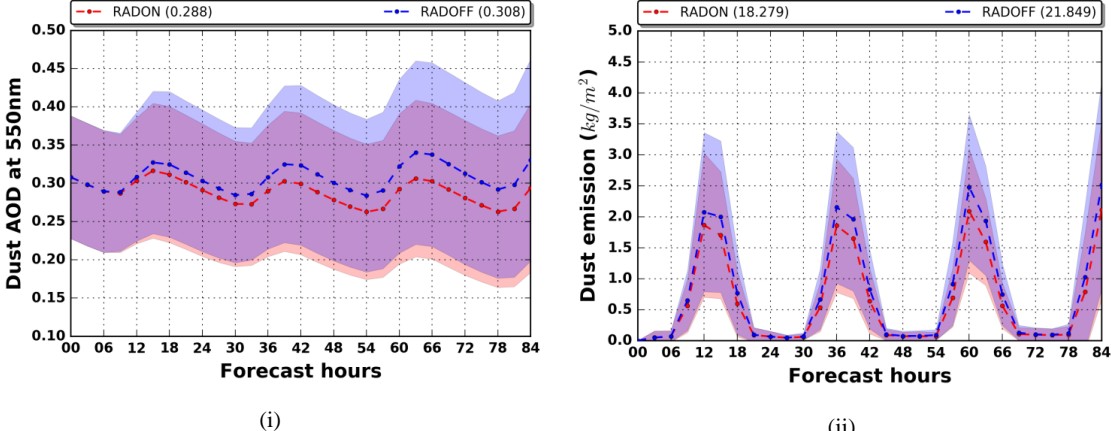

(i)  (ii)

**Figure 9:** (i) Regional dust AOD at 550nm averaged over the simulation domain (NSD) and (ii) Regional dust emission (in kg m$^{-2}$) aggregated over the simulation domain (NSD). Blue and red curves correspond to the mean values, calculated from the 20 desert dust outbreaks, for the RADOFF and RADON simulations, respectively, and the shaded areas represent the associated standard deviation.



**Figure 10:** Timeseries of the downwelling: (i) SW and (ii) LW radiation measured at Sede Boker (red line) and simulated based on the RADON (black line) and RADOFF (blue line) configuration of the NMMB-MONARCH model during the periods: (a) 22 Feb. 2004 00UTC – 25 Feb. 2004 12UTC and (b) 21 Apr. 2007 00UTC – 24 Apr. 2007 12UTC. The mean ground and modelled values along with the computed correlation coefficients (R) between RADON-BSRN and RADOFF-BSRN, both calculated over the simulation periods, are also provided. (iii) Timeseries of the simulated dust AOD at 550 nm for the RADON (black line) and RADOFF (blue line) configuration of the NMMB-MONARCH model. Moreover, the AERONET total AOD at 500 nm (red) and AERONET alpha (green) values are provided.



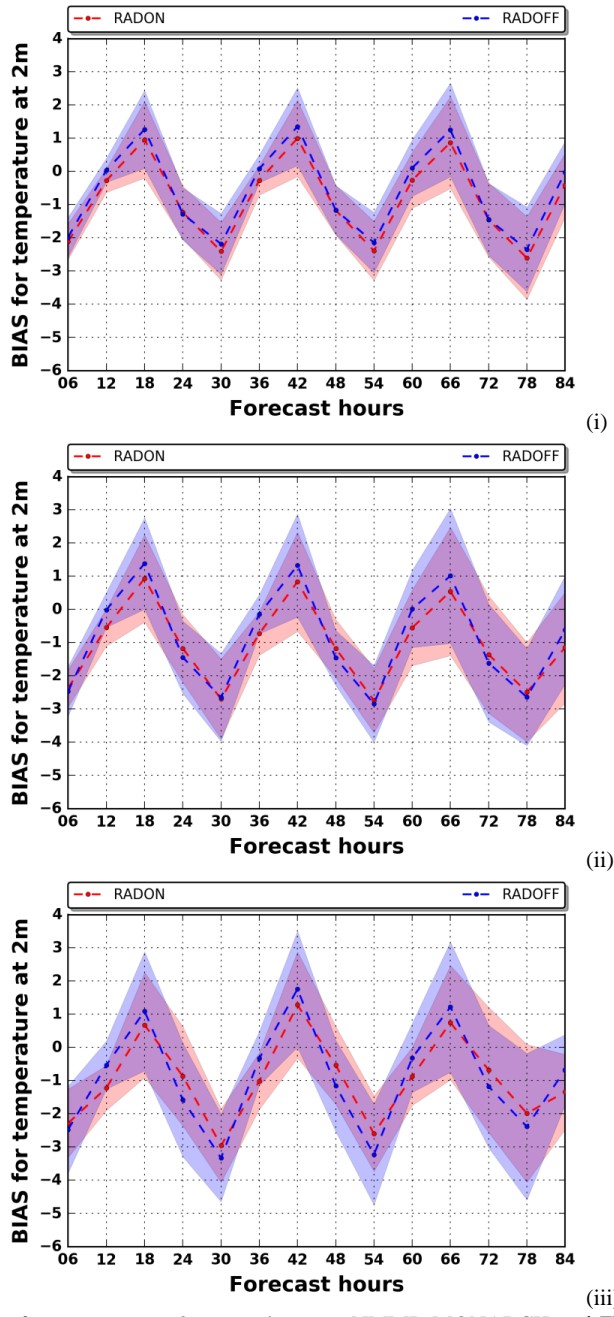

**Figure 11:** Regional biases of temperature at 2 meters between NMMB-MONARCH and FNL, at 1°x1° degrees spatial resolution, calculated over land grid points of the simulation domain (NSD) in which dust AOD at 550 nm is higher/equal than: (i) 0.1, (ii) 0.5 and (iii) 1.0.





**Figure 12:** Vertical profiles of the regional temperature RADON-FNL (red curve) and RADOFF-FNL (black curve) biases calculated over grid points (1° x 1° degrees spatial resolution) where the dust AOD at 550 nm is higher/equal than: (i) 0.1, (ii) 0.5 and (iii) 1.0. In addition, the vertical profiles of the simulated dust concentration (in x10$^{-6}$ kg m$^{-3}$) are provided (brown curve). Each profile corresponds to the mean value calculated from the 20 desert dust outbreaks which are considered while the shaded areas correspond to the associated standard deviations. The obtained results are valid: (a) 24 and (b) 48 hours after the initialization of the forecast period.