# Peer review of "Direct radiative effects during intense Mediterranean desert dust outbreaks"

_Atmospheric Chemistry and Physics, 2017_

## Referee Comment (RC1) · Anonymous Referee #1 · 12 Jan 2018

In my opinion the manuscript Direct radiative effects of intense Mediterranean desert dust outbreaks is acceptable for publication in ACP in its current state.

---

## Referee Comment (RC2) · Anonymous Referee #2 · 18 Jan 2018

The paper addresses an important aspect of the Mediterranean radiation budget and climate. Intense Saharan dusts event may produce large perturbations to radiation, and affect surface temperature, heat exchange at the surface, circulation, etc. The study uses satellite data to identify intense events. Effects on radiation and different processes are investigated for the selected cases using a regional model which includes dust and radiation.

The paper is an interesting and useful contribution to the understanding of dust role and interactions in the Mediterranean.

A couple of aspects may be improved.

The radiative effects are strongly related with the aerosol optical depth (AOD). A com-

parison of AOD values produced by the model versus those obtained from MODIS is presented in the paper. However, the comparison is qualitative and for a selection of cases. Given the large role of AOD in determining the radiative effects, a more detailed, possibly quantitative, comparison should be carried out. On the same point, some reference is made throughout the text to the inability of the model in reproducing the amount of dust. This should be better assessed.

Some results, mainly in the shortwave spectral range, may be linked to differences in the surface albedo, in particular between ocean and land/desert. The discussion of this point may be somewhat improved. In some cases averages over the Mediterranean Satellite Domain (MSD) have been used. The domain includes land and ocean surfaces. I would suggest separating the estimates of radiative effects obtained on land from those obtained over the ocean. Summing/compensation effects, also dependent on the fraction of surface type occurring in each event, may be present when the average includes land and ocean surface types.

Minor points are outlined below.

lines 17-19: please, indicate the AOD range attained during the selected events.

l. 21-26: please, specify for what AOD and over what area these vary large radiative effects are found.

l. 66-68: the sentence is not clear; please, rephrase it

l. 153: I would suggest specifying here that the dust outbreaks are identified using daily multi-sensor satellite data

l. 188-: please, clarify the difference between pixel and grid cell: are those the same?

table 1: are all the selected cases classified as "extreme" events? Are there "strong" events among them? Is there information on the time duration of the events?

l. 308: "quadratic" should be "quadrature"

l. 313: the correct web address seems to be: http://rtweb.aer.com/

l. 317: maybe "fraction" insted of "percentage"

l. 324: it may be useful to add here information on the used refractive indices. They play a central role in the determination of the radiative effects, and the reader should be aware of which set of refractive index values are used in the calculations.

section 5.1: as discussed above, the comparison between satellite and modelled AOD seems qualitative. Given the stated limitations of the satellite dataset over land, a quantitative comparison might be carried out over the ocean. Also, the use of different colour scales in figure 3 does not allow a more detailed comparson.

l. 552: may the differences between the results over the MSD amd SDD domains be partly due to the albedo differences? I would expect an effect, mainly for the NETSURF component.

l. 596: does the model produce substantially different dust size distributions over the Sahara and along the coast and in the Mediterranean? It might be interesting to show this effect.

section 5.3. the dust outbreak impact on SH and LE is investigated only over land. It may be worth including in this section the discussion on the SH and LE changes in the marine environment (l. 751-752). This is also needed to support the validity of the estimated temperature biases over the ocean discussed in section 5.4.

l. 734-: it may be worth recalling the AOD value which correponds with these cross sections.

l. 858: although pyrgeometers are sensitive to the wavelength range 4-50 micron or similar, they are calibrated to provide LW irradiances integrated up to 100 micron.

l. 796-803: how is the dust emission calculated? It should be mainly related to the wind intesity, and it seems to me that such a large day/night difference may be explained only

if the emission is calculated as dust entrainment at some altitude above the ground

section 5.6: the varification of the data against surface radiation measurements is a very ambitious task. As the authors state, it would require a very good model description of the dust event evolution and spatial distribution, and a good reproduction of the observed AOD. I would suggest shortening this section, removing the discussion of specific cases and figure 10, and presenting the results as statistical means for all considered sites (a condensed version of tabel S1). Some of the selected events have been previously investigated using satellite/ground based measurements, and radiation transfer modelling (see e.g., Santese et al., 2010; Benas et al., 2011; di Sarra et al., 2011). The authors may consider if it mey be reasonable to compare the radiative effect estimates, instead of the irradiances, obtained during some of these events.

l. 859: I assume that emission from atmospheric gases and from the surface is not included in the way the SW radiation (up to 12.2. microns) is calculated. This might be clarified.

References

Benas, N., N. Hatzianastassiou, C. Matsoukas, A. Fotiadi, N. Mihalopoulos, and I. Vardavas (2011), Aerosol shortwave direct radiative effect and forcing based on MODIS Level 2 data in the Eastern Mediterranean (Crete), Atmos. Chem. Phys., 11, 12647-12662.

di Sarra, A., C. Di Biagio, D. Meloni, F. Monteleone, G. Pace, S. Pugnaghi, and D. Sferlazzo (2011), Shortwave and longwave radiative effects of the intense Saharan dust event of 25-26 March, 2010, at Lampedusa (Mediterranean sea), J. Geophys. Res., 116, D23209, doi: 10.1029/2011JD016238.

Santese, M., M. R. Perrone, A. S. Zakey, F. De Tomasi, and F. Giorgi (2010), Modeling of Saharan dust outbreaks over the Mediterranean by RegCM3: Case studies, Atmos. Chem. Phys., 10, 133-156.

---

## Referee Comment (RC3) · Anonymous Referee #3 · 9 Feb 2018

The paper presents an interesting study for calculating DRE with the use of the NNMB-MONARCH model (NNMB). It is a well written paper which with the following revisions it could be published in the ACP journal. My main comments are:

- In order to accept the results of such a study, a more comprehensive validation of the presented outputs using real measurements and an analysis of the uncertainties introduces in several phases of the method have to be presented.

- A major aspect of the paper is not clarified. The abstracts talks about DRE and as the authors point out this is mostly aerosol optical depth (AOD) dependent. MODIS retrieves total (dust + other types) AOD while NNMB only dust AOD (that is what is shown throughout the text and in e.g. figure 3). So the authors have to clarify if they talk about Dust DRE or DRE. If someone assumes that these 20 events are purely dust

events, an AOD comparison of MODIS AOD and NNMB AODs have to be included (not quantitevily as in fig. 3), in order try to assess the model results.

- A major issue of the paper is the link between the NNMB results and the Gkikas et al., methodology (GM) for identifying dust episodes. Some questions that have to be clarified on the manuscript are the following:

(i) Are the domains seen in figure 3 and 1 have been used in the GM for all the episodes that are presented in the table 1? Is there a mix of surface and sea Modis pixels used?

(ii) When GM identifies an episode (e.g. example of figure 3) are the DRE calculations of NNMB account only the relative (episodic) modis pixels ? I think the answer here is no but it has to be clarified. So, If the answer is no (thus the whole domain (e.g. MSD) is used for NNMB) then the importance of the GM episode identification is only partially valid. (e.g. a lot of white in fig. 3 are used based only on NNMB and not on GM). As identifying an episode in a limited area in the MSD domain does not mean that this is valid for the whole domain.

(iii) If the whole domain is used are results of table 1 dependent in addition to dust AOD to the spatial extension of the event? Can a number of different episodes with different spatial extends and AODs, averaged (table 2)?

Another example of the last point above is that modis GM detects a plume (high AOD) covering very few pixels in the western part of MSD (for example last row of figure 3). Then based on GM the whole MSD domain is considered as the one that will provide the DRE. In this case the link on GM used as a proxy in this work is very week as it covers only a small part of the domain, plus AODs are not compared. So also a number of episodic pixels should be included in these GM dust episode restrictions. Or simply dust outbreak identification can be based on NNMB spatial and NNMB-AOD absolute criteria as now the link with GM is really weak.

In addition, in this case (and others e.g. west domain of fig. 3b) NNMB dust pixels

cover less than 50% of the MSD. When averaging the 20 cases this percentage of pixels varies a lot. In the end you are averaging and provide a result e.g. SW = -9.7. So some of the outbreaks contribute much more and some others not, based on the dust coverage on the MSD only. Where can such statistics be used?

To summarize, if GM is not used for AOD validation and GM identifies as "dust episodic pixels" only a fraction of the pixels used finally from NNMB for calculating dust DRE, then its use becomes not important for this study. So if someone trusts NNMB for DRE calculations, then it is much more easy to trust it also for dust outbreak identification.

- There are more than 100 references and a lot of discussion about aerosol effects and model applications, but very few about NNMB validation on e.g. AOD retrievals. And only one (Ohmura) on BSRN radiation related validation. I think it is more essential to prove the validity of AOD NNMB output (e.g. radiation) and intermediate parameters (e.g. AOD), than a numerous studies cited here, with a very theoretical link to the paper.

- The validation using BSRN is incomplete. In the document and in the abstract you are talking about this validation and 8 stations. Then in the manuscript only one station is shown. And from that only 4 days. In order to validate the results a more comprehensive analysis of long term periods of these 8 stations is needed. Probably Ohmura has answered some of the validation related questions, but this paper focuses on "intense dust outbreaks", and a specific model, so results might differ from the Ohmura related ones.

- There are several issues that have to be clarified/commented on the input parameters of the model:

(i) Optical properties proposed in figure 2. Have been validated ?

(ii) Water vapor, carbon dioxide, ozone, methane and oxygen. Where do you find these inputs ?

(iii) Differences in dust optical properties of Sahara and middle East sources What did you use and how much uncertain are they? and what is the contribution of this uncertainty in the final DRE budget ?

- BSRN and model differences in wavelength integrals of solar radiation. You mention: "These differences might contribute to the level of agreement between model and observations; however, are not discussed in our evaluation analysis". I think this is an important issue that have to be clearly discussed if a proper validation is included.

- As already mentioned AOD comparisons from MODIS and NNMB could add value to this work. "The model's ability to reproduce correctly the spatial patterns and values of dust AODs is crucial for a successful computation of the dust DREs, since DREs are determined to a large extent by AOD". In addition you are mentioning modis uncertainty in section 2. Is this getting high (e.g. ∼0.5) for both sea and mostly surface retrievals when you examine AODs in the order of 2-3 based on the table 1? And is this uncertainty already important for such outbreaks for the GM and indirectly for the DRE related uncertainty ?

- Table 2. These statistics are not referring to the model uncertainty but is an averaging of the episodes provided by the GM. NNMB DRE uncertainty is much more useful for any future user of these results. For example a systematic bias can not be identified here. This is also because the GM thresholds are mostly subjective as:

(i) Mean AOD values on dust related areas do not have an important statistical meaning due to the non normal distribution of AOD. It is clear that this is a published work and I have tried to follow the previous work by Gkikas et al and the relative open discussion, describing the method. However, as this is an open public statement I have to comment that AOD does not follow necessarily a normal distribution so using the mean is not absolutely correct. Moreover, dust outbreaks related pixels/locations can be characterized more from a bimodal distribution of AODs when another (than dust) important AOD source is rarely present (e.g. most of the marine grids of Mediterranean

domain).

(ii) GM: By definition high mean AOD values per pixel are closer to dust sources. That makes possible that a pixel with high (in an absolute sense) AOD close to a dust source to be considered non episodic and a pixel with lower AOD, away from the sources to be considered episodic. This is ok, as it is just a matter of definition. But it gets more important when it is used for DRE calculations. So, the latest can be problematic when you calculate DRE in dust outbreaks or filter the outbreaks, as for the first example pixel (high AOD) it is not an outbreak and for the second (lower AOD) it is characterized as an outbreak. The results using this method for DRE calculations become not easily useful and applicable.

- Last but very important, the paper is very long and in various cases the discussion includes a lot of details that in the end confuse the reader on what is the important findings here and which are not. Even for scientists in the field it becomes difficult to read. Authors have to try to reduce the length of the manuscript keeping the important aspects of the results presented. Basically for section 5 I would suggest to try to take out a lot of information that are secondary and to focus on the important results.

Minor comments:

Line 141: it has already mentioned previously.

Line 173: developed – improved

Table 1: episodes = grid cells

The overall approach of this paper is valuable and worth publishing. I strongly believe that after the above revisions, corrections and additional analysis it will be essentially upgraded and then it could be published in the ACP journal.

---

## Author Comment (AC1) · 15 May 2018

**Response to Reviewer1**

**"In my opinion the manuscript Direct radiative effects of intense Mediterranean desert dust outbreaks is acceptable for publication in ACP in its current state."**

Thanks!

Before our paper being published in ACPD, the reviewer made the following comment and the Editor suggested that it should be answered in the current review process. Please find our response (regular font) below the reviewer's comment (bold font).

**"My major concern is how the model takes into account the relative humidity to scale the optical (e.g. real and imaginary refractive indices) and the microphysical properties (e.g. size) of the aerosols. As authors well-know and say within the manuscript the water vapor influences SW and LW spectral ranges. It is not clear for me how the RRTMG considers this significant aspect (RH). An analysis the relative humidity (RH) in this area for these 20 desert dust cases would be very clarifying because it is huge important to go through how the RH changes from day to night times and how the desert dust optical and microphysical properties vary from day to night times. Relevant parameters in your calculations are the mass extinction efficiency, the single scattering albedo and the asymmetry parameter. The effect of the RH over them is well explained by Myhre et at. (1998)."**

In the NMMB-MONARCH model, dust aerosols are externally mixed and hydrophobic. Therefore, no hygroscopic growth is considered and subsequently the RH effects are not taken into account in the RRTMG. This assumption, it is not expected to introduce large errors since it is well documented in literature that mineral particles are mainly hydrophobic and consisted of insoluble substances, particularly over desert regions. Of course, it is also known (e.g. Sullivan et al., 2009; Knippertz and Stuut, 2014) that dust hygroscopicity increases through mixing soluble of hygroscopic material with insoluble mineral particles, thus leading to the formation of internal mixtures of dust and sulfate, which can make mineral particles more soluble. Nevertheless, it should be noted that for such atmospheric processing to take place, time is needed and that this increase of dust hygroscopicity mainly occurs through aging. However, our study focuses on intense dust episodes above the Mediterranean, which basically transport fresh, and thus hygrophobic, dust particles. This clarification has been added in the revised manuscript (Lines 283-284). The paper of Myhre et al. (1998), regarding the effect of RH on optical properties, refers to sulfate and soot aerosols, and not dust.

---

## Author Comment (AC2) · 15 May 2018

**Response to Reviewer2**

We would like to thank the Reviewer for the useful comments that helped us to improve our manuscript. Below are given point by point answers to the comments (also provided in bold font).

**"The paper addresses an important aspect of the Mediterranean radiation budget and climate. Intense Saharan dusts event may produce large perturbations to radiation, and affect surface temperature, heat exchange at the surface, circulation, etc. The study uses satellite data to identify intense events. Effects on radiation and different processes are investigated for the selected cases using a regional model which includes dust and radiation.**

**The paper is an interesting and useful contribution to the understanding of dust role and interactions in the Mediterranean.**

**A couple of aspects may be improved."**

**"The radiative effects are strongly related with the aerosol optical depth (AOD). A comparison of AOD values produced by the model versus those obtained from MODIS is presented in the paper. However, the comparison is qualitative and for a selection of cases. Given the large role of AOD in determining the radiative effects, a more detailed, possibly quantitative, comparison should be carried out. On the same point, some reference is made throughout the text to the inability of the model in reproducing the amount of dust. This should be better assessed."**

As suggested by the Reviewer, we have made a more detailed comparison between the observed (MODIS) and simulated (NMMB) AODs. In order to eliminate the spatial inconsistencies between the two products, we have regridded the model outputs from their raw spatial resolution (0.25° x 0.25°) to 1° x 1° in order to match them the resolution of satellite retrievals. The new geographical distributions of the modelled AODs (dynamically calculated dust plus GOCART climatology for the other aerosol types), at coarse spatial resolution, have replaced the old ones presented in Figures 3 and S1 of the previous version of manuscript. In both MODIS and NMMB patterns a common colorbar is used making easier a visual intercomparison for the reader. Moreover, the model AODs have been compared against those of MODIS, considering only the grid cells where a DD episode (either strong or extreme) has been identified by the satellite algorithm. Note that NMMB-MODIS comparison all over the MSD is not possible because of the gaps (white areas) in MODIS AOD distributions, given that the operation of MODIS retrieval algorithm is impossible therein. The obtained results for each episode, in terms of overall computed correlation coefficient and bias (defined as NMMB-MODIS) are given in Fig. R1, while the stacked bars illustrate the number of strong, extreme and total DD episodes for each case (available also in Table 1).

Among the studied cases, it is revealed a strong variation of R values (Figure R1-ii) which reflects the diversity of the model's capability in terms of capturing the spatial patterns of the desert dust outbreaks. These drawbacks rise mainly from displacements of the simulated dust patterns with respect to the observed ones (see Figs 3 and S1). The best performance is found on 22 Feb 2004 (R=0.82) while in 7 out of 20 cases R values are higher than 0.5. As it concerns the bias, in absolute terms, in all the events negative values are recorded ranging from -2.3 (24 Feb 2006) to -0.17 (19 May 2008). This finding shows that the model underestimates consistently the intensity of the desert dust outbreaks which have been analyzed in the present study.

According to the evaluation analysis, the model's ability in terms of reproducing satisfactorily the dust fields varies strongly case-by-case while the simulated intensity of the desert dust outbreaks is lower with respect to the satellite retrievals. It should be noted that the level of agreement between observed and simulated AODs (Lines 451-465) is not only associated with the model deficiencies, but also with other factors like the temporal inconsistency between the two products. More specifically, the satellite retrievals correspond to daily averages whereas the model products are

representative for a specific forecast time (instantaneous fields). Considering the high variability of aerosols' loads, particularly under episodic conditions, this temporal discrepancy imposes a limitation when a quantitative comparison between MODIS and NMMB is attempted. This fact can explain the observed differences found either on the intensity or on the spatial patterns of the desert dust events. Also, it must be considered that artifacts of the satellite retrievals (e.g. clouds contamination, representativeness/homogeneity within the 1° x 1° grid cell) may lead to higher AODs as it has been shown in relevant evaluation studies (Gkikas et al., 2016). In the revised manuscript, the discussion in Section 5.1 has been updated presenting the quantitative comparison of NMMB-MONARCH versus MODIS-Terra as well as the reasons which lead to deviations between these two products.

Finally, we would like to bring to the attention of the Reviewer that a detailed evaluation of the same version of the NMMB model for 2006 has been presented in Pérez et al. (2011), who compared the model products against MISR and AERONET retrievals. Based on their findings, for a domain including the Mediterranean, it is revealed that the model in general is able to reproduce satisfactorily the spatiotemporal features of the desert dust fields. Moreover, an evaluation of the NMMB AOD forecasts, along with similar forecasts from other models, against ground based AERONET and satellite MODIS retrievals, is available at the weblink of SDS-WAS System (https://sds-was.aemet.es/forecast-products/forecast-evaluation) to which reference is now made in the revised manuscript (lines 303-306).

[Figure]

[Figure]

(iii)

**Figure R1:** (i) Number of strong (green bars), extreme (red bars) and total (entire bars) DD episodes identified by the satellite algorithm, (ii) Correlation coefficients (R) between satellite and model AODs, (iii) Regional average biases between the NMMB simulated and the MODIS retrieved AODs. Results are given for each studied case (given in x-axis) and are computed taking into account only pixels over which a DD episode (either strong or extreme) has been identified by the satellite algorithm.

**"Some results, mainly in the shortwave spectral range, may be linked to differences in the surface albedo, in particular between ocean and land/desert. The discussion of this point may be somewhat improved. In some cases, averages over the Mediterranean Satellite Domain (MSD) have been used. The domain includes land and ocean surfaces. I would suggest separating the estimates of radiative effects obtained on land from those obtained over the ocean. Summing/compensation effects, also dependent on the fraction of surface type occurring in each event, may be present when the average includes land and ocean surface types."**

The regional SW DREs for the MSD have been calculated separately over land and sea and the obtained results are illustrated in Figure R2. The temporal variation of SW $DRE_{ATM}$, $DRE_{SURF}$ and $DRE_{NETSURF}$ values is similar with the one presented for the whole Mediterranean domain (Figure 5 in the revised document) over both land and ocean areas. However, a careful eye look reveals differences between land and ocean DREs. Thus, over dark (sea) surfaces $DRE_{SURF}$ and $DRE_{NETSURF}$ values are almost equal (Fig. R2-ii) while over brighter (land) surfaces $DRE_{NETSURF}$ values clearly differ by $DRE_{SURF}$ ones, i.e. they are smaller, due to the higher surface albedo, leading to increasing upward component and reducing the absorbed radiation. Another difference between land and ocean DREs is the larger magnitude of surface DREs over ocean than land areas, especially in early forecast times, due to higher AODs over ocean. The most noticeable difference between the Mediterranean land and sea DREs is evident at TOA, both in terms of temporal variation and magnitude, clearly reflecting the role of the surface albedo. In particular, over land, the $DRE_{TOA}$ values are maximum (up to 9 $Wm^{-2}$) during early morning and afternoon hours, decreasing in magnitude between 9-12 UTC (values ranging from -3.6 to -2.2 $Wm^{-2}$) while such a decrease is not observed over sea areas. Also, the magnitude of ocean $DRE_{TOA}$ values is smaller than over land, i.e. a stronger cooling of the Earth-atmosphere system is produced by aerosols over oceans than land due to the low sea water albedo below aerosols. The overall computed SW DREs presented in Figure 5 (without discriminating between land and sea grid points of the NMMB-MONARCH model) are mainly driven by the corresponding DREs over continental Mediterranean areas. The aforementioned result is also valid for the whole simulation domain (NSD) as well as for the Sahara domain (SDD). In the revised document a short sentence has been added (Lines 582-584).

[Figure]

**Figure R2:** Regional all-sky SW DREs calculated over the MSD above: (i) land and (ii) sea areas.

Minor points are outlined below.

**"lines 17-19: please, indicate the AOD range attained during the selected events."**

We have added in the text (lines 22-23) the range of the maximum dust AODs (2.5 – 5.5) simulated by the NMMB-MONARCH model.

**"l.21-26: please, specify for what AOD and over what area these vary large radiative effects are found."**

Done. Please see Lines 23-31.

**"l. 66-68: the sentence is not clear; please, rephrase it"**

The following sentence in the submitted document has been replaced with a new one (written below) in the revised text (Lines 73-76).

**OLD (submitted manuscript)**
"Through this chain of complex processes, it is described the indirect impact of mineral particles on the radiation and compared to the other two dust radiative effects (direct and semi-direct) is characterized by even larger uncertainties."

**NEW (revised manuscript)**
"This chain of complex processes, involving aerosol-cloud-interactions (ACI) and the subsequent modifications of the radiation fields, constitute the indirect impact of mineral particles on radiation, which is characterized by the largest uncertainties, even larger than those of the dust direct and semi-direct effects."

**"l. 153: I would suggest specifying here that the dust outbreaks are identified using daily multi-sensor satellite data"**

Done. Please see Lines 160-163 in the revised manuscript.

**"l. 188-: please, clarify the difference between pixel and grid cell: are those the same?"**

Both terms have the same meaning. In order to be clear we have added this clarification in Lines 200-202.

**"table 1: are all the selected cases classified as "extreme" events? Are there "strong" events among them? Is there information on the time duration of the events?"**

For each dust outbreak there are pixel-level episodes that are either strong or extreme, according to their AOD values. As suggested by the Referee, we have added in Table 1 two columns giving the number of strong and extreme DD episodes for each dust outbreak. Moreover, in the revised manuscript we have included this information by providing the ranges for the strong and extreme DD episodes that took place within the MSD (see Lines 224-228). No information is given about the duration of studied events because according to our analysis, the maximum duration (consecutive days satisfying the defined criteria, see sect. 2) is two (2) days, but in such cases we have decided to keep just the day for which the number of total pixel-level DD episodes is higher (see Lines 216-217).

| Case | Date | Strong DD episodes | Extreme DD episodes | Total DD episodes | Intensity | Affected parts of the Mediterranean domain |
|---|---|---|---|---|---|---|
| 1 | 31 July 2001 | 56 | 29 | 85 | 0.74 | Western |
| 2 | 8 May 2002 | 20 | 51 | 71 | 1.60 | Central |
| 3 | 4 April 2003 | 23 | 30 | 53 | 1.42 | Eastern |
| 4 | 16 July 2003 | 38 | 45 | 83 | 0.98 | Western and Central |
| 5 | 22 February 2004 | 10 | 36 | 46 | 2.18 | Central and Eastern |
| 6 | 26 March 2004 | 28 | 38 | 66 | 1.45 | Central and Eastern |
| 7 | 27 January 2005 | 12 | 25 | 37 | 1.36 | Central and Eastern |
| 8 | 2 March 2005 | 8 | 37 | 45 | 2.96 | Central and Eastern |
| 9 | 28 July 2005 | 10 | 20 | 30 | 1.08 | Western and Central |
| 10 | 24 February 2006 | 3 | 42 | 45 | 2.92 | Eastern |
| 11 | 19 March 2006 | 11 | 28 | 39 | 1.37 | Eastern |
| 12 | 24 February 2007 | 8 | 34 | 42 | 2.29 | Central and Eastern |
| 13 | 21 April 2007 | 15 | 27 | 42 | 1.65 | Central |
| 14 | 29 May 2007 | 17 | 30 | 47 | 1.40 | Eastern |
| 15 | 10 April 2008 | 9 | 33 | 42 | 1.58 | Central |
| 16 | 19 May 2008 | 16 | 50 | 66 | 1.45 | Central |
| 17 | 23 January 2009 | 4 | 32 | 36 | 2.65 | Eastern |
| 18 | 6 March 2009 | 18 | 23 | 41 | 1.41 | Eastern |
| 19 | 27 March 2010 | 10 | 29 | 39 | 1.43 | Central |
| 20 | 2 August 2012 | 12 | 23 | 35 | 1.20 | Western |

**"l. 308: "quadratic" should be "quadrature""**

We have corrected it.

**"l. 313: the correct web address seems to be: http://rtweb.aer.com/"**

We have corrected this. Thanks for the note.

**"l. 317: maybe "fraction" instead of "percentage""**

We have changed the text according to the reviewer's suggestion.

**"l. 324: it may be useful to add here information on the used refractive indices. They play a central role in the determination of the radiative effects, and the reader should be aware of which set of refractive index values are used in the calculations."**

In Lines 331-336 of the revised manuscript, we have provided information on the refractive indices used in the model, as requested by the reviewer. More specifically, it is now specified that the

refractive indices used in our simulations were taken by GADS (Koepke et al., 1997) and modified following Sinyuk et al. (2003), as described in Pérez et al. (2011).

**"section 5.1: as discussed above, the comparison between satellite and modelled AOD seems qualitative. Given the stated limitations of the satellite dataset over land, a quantitative comparison might be carried out over the ocean. Also, the use of different colour scales in figure 3 does not allow a more detailed comparison."**

Please see our response to your first main comment.

**"l. 552: may the differences between the results over the MSD and SDD domains be partly due to the albedo differences? I would expect an effect, mainly for the NETSURF component."**

In the shortwave spectrum, the surface albedo plays a critical role on the observed differences between the calculated DREs in the MSD and SDD. This is evident at noon when positive (planetary warming) and negative (planetary cooling) $DRE_{TOA}$ values are found over the Sahara and the Mediterranean, respectively (Figure 5). In the former region, due to the higher surface albedo the atmospheric warming enhances (mineral particles do not absorb only the incoming SW radiation but also the reflected radiation from the ground) dominating over the surface (NETSURF) cooling which decreases since the upward component (reflected radiation from the ground) increases. On the contrary, over dark areas (maritime environments or vegetated land) the dust layers are brighter than the underlying surface resulting in negative perturbations (cooling effect) at TOA. Summarizing, the contrast between low- and high-reflective surfaces doesn't affect only the absorbed radiation at the ground (NETSURF) but also the atmospheric radiation budget and subsequently the perturbation of the Earth-Atmosphere system's radiation budget (Eq. 4).

**"l. 596: does the model produce substantially different dust size distributions over the Sahara and along the coast and in the Mediterranean? It might be interesting to show this effect."**

In Figure R3 is depicted the geographical distribution of the coarse-to-fine ratio of dust aerosols, at 12 UTC on 2nd August 2012, which has been calculated by dividing the aggregated dust concentrations for bins 5-8 (coarse particles) and bins 1-4 (fine particles). As expected, the maximum ratios (~ 19) are found over/close the dust sources (central Algeria) whereas considerably high values (> 10) are observed in the western parts of Sahara and over the Atlantic Ocean, both affected by the major dust plume (see Figure 4 in the manuscript).

[Figure]

**Figure R3:** Geographical distribution of coarse-to-fine ratio of dust aerosols at 12 UTC on 2nd August 2012.

**"section 5.3. the dust outbreak impact on SH and LE is investigated only over land. It may be worth including in this section the discussion on the SH and LE changes in the marine**

**environment (l. 751-752). This is also needed to support the validity of the estimated temperature biases over the ocean discussed in section 5.4."**

As stated in lines 664-665, in the utilized version of NMMB-MONARCH model the atmospheric driver is not coupled with an ocean model. Therefore, not a significant impact on SH and LE is expected over maritime areas, since the feedbacks from ocean are neglected. In addition, due to the larger heat capacity of sea (Lines 736 - 743), the perturbations of the SH and LE fields should be negligible at short temporal scales. The aforementioned reasons explain why we have investigated the induced impacts on heat fluxes (Section 5.3) only over land areas.

**"l. 734-: it may be worth recalling the AOD value which corresponds with these cross sections."**

We have inserted in the text the maximum dust AODs at 550nm simulated by the NMMB-MONARCH model along the cross-sections (see lines 732-733, lines 750-753).

**"l. 858: although pyrgeometers are sensitive to the wavelength range 4-50 micron or similar, they are calibrated to provide LW irradiances integrated up to 100 micron."**

We have taken this information from the specifications of the pyrgeometers (Eppley-PIR and Kipp & Zonen CGR4) that are installed at BSRN stations. The spectral ranges of the measured downwelling LW radiation at the ground span the wavelength range from 4 to 50 microns for Eppley-PIR and from 4.5 to 42 microns for Kipp & Zonen CGR4. Nevertheless, we haven't found any relevant reference regarding the calibration procedure that extends the upper bound to 100 microns.

**"l. 796-803: how is the dust emission calculated? It should be mainly related to the wind intensity, and it seems to me that such a large day/night difference may be explained only if the emission is calculated as dust entrainment at some altitude above the ground"**

We have avoided in our paper to provide much information about the dust emission scheme since a detailed description is given by Pérez et al. (2011). Briefly, the saltation of mineral particles is approximately proportional to the third power of the wind speed. The vertical dust flux ($F_k$), constrained by a tuning factor, is proportional to the horizontal flux. Based on $F_k$ and turbulent regime, the concentration of the emitted dust particles is diagnosed at the top of a viscous sublayer extending between the assumed smooth desert surface and the lowest model layer. During day, due to thermal convection, the turbulence is enhanced resulting thus to an unstable atmosphere, higher wind speeds and subsequently to larger amounts of emitted dust. On the contrary, during night, the atmosphere is more stratified (less turbulence) leading to weaker wind speeds and less dust emission. The strong variability of dust emission throughout the day, presented in Figure 6-ii of our manuscript, has been also reported in previous studies (e.g. Schepanski et al., 2009).

**"section 5.6: the verification of the data against surface radiation measurements is a very ambitious task. As the authors state, it would require a very good model description of the dust event evolution and spatial distribution, and a good reproduction of the observed AOD. I would suggest shortening this section, removing the discussion of specific cases and figure 10, and presenting the results as statistical means for all considered sites (a condensed version of table S1). Some of the selected events have been previously investigated using satellite/ground based measurements, and radiation transfer modelling (see e.g., Santese et al., 2010; Benas et al., 2011; di Sarra et al., 2011). The authors may consider if it may be reasonable to compare the radiativeeffect estimates, instead of the irradiances, obtained during some of these events."**

We would like to remind that the goal of this study is not to evaluate the model's radiative fluxes against measurements, but to highlight the model improvement in terms of more adequately

reproducing radiative fluxes when it takes into account dust in its simulations. Thus, we prefer to keep Figure 10 and the relevant discussion, since in both example cases (in Sede Boker) is nicely depicted (highlighted) the role of factors affecting the level of agreement between NMMB and BSRN, by taking advantage of the existing concurrent AERONET retrievals while the impact of clouds (relied on numerical simulations) is also considered. It is the first time that such an evaluation analysis of the NMMB-MONARCH is presented.

Regarding the last part of the reviewer's comment, following his suggestion, we have compared our SW DREs with the corresponding ones calculated in Benas et al. (2011) and the results are presented in Table R1. The surface DREs (SURF, NETSURF) are comparable but lower (by up to 12 Wm$^{-2}$ and 8 Wm$^{-2}$, respectively) in our analysis while the atmospheric warming in Benas et al. (2011) is 2.6 times higher than ours. At TOA, our SW DRE reach down to -35 Wm$^{-2}$, being higher, in absolute terms, by 59% with respect to Benas et al. (2011). A significant difference between the two studies, determining the DRE calculations, is that in our case the AOD (0.09) and SSA (0.87) are very low in contrast to Benas et al. (2011) where the corresponding values are equal to 0.44 and 0.95, respectively. Therefore, higher loads are considered in Benas et al. (2011) whereas the suspended particles are more absorptive in our analysis. Both facts interpret the differences found between the two studies. An additional source of differences is that DREs in our calculations are representative 60 hours after the initialization of the model (00 UTC 24-Feb-2006) while they have been spatially averaged around the FORTH-CRETE AERONET station (Latitude: 35º-36º N, Longitude: 25º-26º E). The increasing errors for increasing forecast time, as well as spatially averaged NMMB DREs against almost local (MODIS' nadir view 10 x 10 km spatial resolution) estimates of DREs in Benas et al. produce differences when comparing our model to Benas et al. (2011) DREs.

In di Sarra et al. (2011), the SW DREs are presented for 25[th] and 26[th] March 2010 while in our study case the forecast run starts at 00 UTC on 27[th] March 2010.

In Santese et al. (2010), the daily averages of DREs are presented for 17[th] July 2003. In the revised supplement document, we are providing the corresponding instantaneous (noon and night) DREs for the same date in Figure S6 (third and fourth row).

**Table R1:** SW DREs at 11:25 UTC on 26-Feb-2006 (Benas et al. (2011)) and at 12 UTC on 26-Feb-2006 (present analysis) over the FORTH-CRETE AERONET station (Crete, southern Greece).

|  | **Benas et al. (2011) [11:25 UTC]** | **Present study [12:00 UTC]** |
|---|---|---|
| **TOA** | -22 Wm$^{-2}$ | -35 Wm$^{-2}$ |
| **SURF** | -66 Wm$^{-2}$ | -54 Wm$^{-2}$ |
| **NETSURF** | -56 Wm$^{-2}$ | -48 Wm$^{-2}$ |
| **ATM** | 34 Wm$^{-2}$ | 13 Wm$^{-2}$ |

**"l. 859: I assume that emission from atmospheric gases and from the surface is not included in the way the SW radiation (up to 12.2. microns) is calculated. This might be clarified."**

In the existing version of the NMMB-MONARCH model, only greenhouse gases and not the emitted short lived atmospheric gases are taken into account. We have added the relevant information in the text (Lines 267-268).

**References**

**Benas, N., N. Hatzianastassiou, C. Matsoukas, A. Fotiadi, N. Mihalopoulos, and I. Vardavas (2011), Aerosol shortwave direct radiative effect and forcing based on MODIS Level 2 data in the Eastern Mediterranean (Crete), Atmos. Chem. Phys., 11, 12647-12662.**

di Sarra, A., C. Di Biagio, D. Meloni, F. Monteleone, G. Pace, S. Pugnaghi, and D. Sferlazzo (2011), Shortwave and longwave radiative effects of the intense Saharan dust event of 25-26 March, 2010, at Lampedusa (Mediterranean Sea), J. Geophys. Res., 116, D23209, doi: 10.1029/2011JD016238.

Santese, M., M. R. Perrone, A. S. Zakey, F. De Tomasi, and F. Giorgi (2010), Modeling of Saharan dust outbreaks over the Mediterranean by RegCM3: Case studies, Atmos. Chem. Phys., 10, 133-156.

---

## Author Comment (AC3) · 15 May 2018

**Response to Reviewer3**

We would like to thank the Reviewer who helped us to improve our paper through his/her report. Below are listed our detailed responses (regular font) to each comment raised by the Reviewer (bold font).

**The paper presents an interesting study for calculating DRE with the use of the NNMB-MONARCH model (NNMB). It is a well written paper which with the following revisions it could be published in the ACP journal. My main comments are:**

**- In order to accept the results of such a study, a more comprehensive validation of the presented outputs using real measurements and an analysis of the uncertainties introduces in several phases of the method have to be presented.**

In the revised manuscript we have made a more detailed comparison between MODIS-NMMB for the cases (dust outbreaks) which are analyzed here. Regarding the validation of radiation and temperature fields, the discussion has been updated whenever is needed. Please see our responses to your comments below.

**- A major aspect of the paper is not clarified. The abstracts talks about DRE and as the authors point out this is mostly aerosol optical depth (AOD) dependent. MODIS retrieves total (dust + other types) AOD while NNMB only dust AOD (that is what is shown throughout the text and in e.g. figure 3). So the authors have to clarify if they talk about Dust DRE or DRE. If someone assumes that these 20 events are purely dust events, an AOD comparison of MODIS AOD and NNMB AODs have to be included (not quantitevily as in fig. 3), in order try to assess the model results.**

We have changed the title of our paper from "*Direct radiative effects of intense Mediterranean desert dust outbreaks*" to "*Direct radiative effects during intense Mediterranean desert dust outbreaks*" so that the goal of our study is more clear. This modification has been made since based on the configuration of the model the amount of dust aerosols is simulated dynamically (online) while for the other aerosol types the GOCART climatology is used (Lines 340-342). Moreover, the DREs are computed for days in which intense dust outbreaks prevail over the greater Mediterranean basin. Under such conditions, and over places where Saharan dust is transported, dust predominates and is the main contributor of AOD, even in MODIS AOD retrievals. Of course, in such cases all aerosol types exert a perturbation of the radiation budget, but the impact of mineral particles is predominant. A quantitative comparison between MODIS and NMMB has been made (suggested also by the Reviewer 2) and the obtained results are presented in Figure S2 (supplementary material) and discussed in Section 5.1.

**- A major issue of the paper is the link between the NNMB results and the Gkikas et al., methodology (GM) for identifying dust episodes. Some questions that have to be clarified on the manuscript are the following:**

**(i) Are the domains seen in figure 3 and 1 have been used in the GM for all the episodes that are presented in the table 1? Is there a mix of surface and sea Modis pixels used?**

The identification of DD episodes through the implementation of the satellite algorithm is made only for the Mediterranean Satellite Domain (MSD, red rectangle in Figure 1) as stated in the manuscript (see lines 192-194). The structure, methodology, and operational phases of the satellite algorithm have been presented in detail by Gkikas et al. (2013, https://www.atmos-chem-phys.net/13/12135/2013/). Briefly, the algorithm operates separately over land and sea surfaces by taking into account the MODIS AODs obtained by the dark target land and ocean retrieval algorithms.

Therefore, the number of DD episodes presented in Table 1 corresponds to the number of grid cells (1° x 1° spatial resolution) where a desert dust (DD) episode has been recorded/identified within the geographical limits of the MSD. Please see lines 207-210 and the caption of Table 1 in the revised manuscript.

**(ii) When GM identifies an episode (e.g. example of figure 3) are the DRE calculations of NNMB account only the relative (episodic) modis pixels? I think the answer here is no but it has to be clarified. So, If the answer is no (thus the whole domain (e.g. MSD) is used for NNMB) then the importance of the GM episode identification is only partially valid. (e.g. a lot of white in fig. 3 are used based only on NNMB and not on GM). As identifying an episode in a limited area in the MSD domain does not mean that this is valid for the whole domain.**

We think that it is clear that the NMMB DREs calculations are made all over the Mediterranean basin and not only over the episodic MODIS pixels. This does not limit the validity and importance of GM dust episode identification. It is self evident that when talking about a dust episode over the Mediterranean, not the entire basin but just a significant part of it is expected to be dominated by dust, which is adequately ensured by GM. Therefore, having "a lot of white in Fig. 3" is not strange, unreasonable or problematic, but on the contrary it is expected and sound. Nevertheless, this does not prevent us from talking about Mediterranean dust episodes and radiative effects (DREs). The only issue that might be relevant to this comment, is averaging regionally over the Mediterranean, where dust and no dust dominated areas are considered together, but even in such cases DRE computations are meaningful. In the revised manuscript, the calculations of the regional DREs have been made taking into account all the grid points and therefore the spatial representativeness is consistent at each forecast step and among the studied cases.

**(iii) If the whole domain is used are results of table 1 dependent in addition to dust AOD to the spatial extension of the event? Can a number of different episodes with different spatial extends and AODs, averaged (table 2)**

In the revised manuscript (Table 1, lines 213-218) it is explained that the frequency and regional intensity, i.e. AOD of 20 dust outbreaks, is calculated from the total pixel-level DD episodes, therefore the results of Table 1, more specifically the intensity, are not dependent on the spatial extend of the episodes. As already answered in the previous comment, regional DREs can be computed for every dust episode. Therefore, as it concerns the second part (sentence) of this comment (e.g. Table 2 results) we believe that averaging DREs over the 20 different dust episodes is meaningful and representative of DREs during Mediterranean dust outbreaks.

**Another example of the last point above is that modis GM detects a plume (high AOD) covering very few pixels in the western part of MSD (for example last row of figure 3). Then based on GM the whole MSD domain is considered as the one that will provide the DRE. In this case the link on GM used as a proxy in this work is very week as it covers only a small part of the domain, plus AODs are not compared. So also a number of episodic pixels should be included in these GM dust episode restrictions. Or simply dust outbreak identification can be based on NNMB spatial and NNMB-AOD absolute criteria as now the link with GM is really weak.**

Most of the content of this comment has already been answered. However, we would like to note that of course intense dust outbreaks are not supposed to cover the entire Mediterranean, on the contrary, they always cover a part of it, this is logical. However, this does not prevent us of talking about dust episodic days over the Mediterranean basin whenever such dust outbreaks occur. And, moreover, it also does not prevent us from computing DREs all over the Mediterranean basin, even averaging over it. Therefore, there is not any problematic link in our concept and methodology combining the

detection of dust outbreaks with GM and the DRE computation with NMMB. Concerning the last sentence and suggestion of the Referee, of course this is an option, i.e. dealing with detection of dust outbreaks and computing the associated DREs solely using the NMMB model. However, this would be purely theoretical. On the opposite, detecting intense dust outbreaks based on an observational approach, i.e. using MODIS products, is more appropriate. Finally, as already stated in our responses to this Referee's previous comments, a comparison of AODs has been made and it is discussed in the revised version of the manuscript.

**In addition, in this case (and others e.g. west domain of fig. 3b) NNMB dust pixels cover less than 50% of the MSD. When averaging the 20 cases this percentage of pixels varies a lot. In the end you are averaging and provide a result e.g. SW = -9.7. So some of the outbreaks contribute much more and some others not, based on the dust coverage on the MSD only. Where can such statistics be used?**

First, we would like to state that in the revised manuscript the regional DREs have been calculated considering all the grid points without setting any criterion on the simulated dust AOD or on clouds (this approach was initially followed). Therefore, at each forecast step and among the 20 desert dust outbreaks the number of grid points is constant. This ensures that the spatial representativeness of the regional DREs does not vary in time and among the studied cases (Figure 5).

**To summarize, if GM is not used for AOD validation and GM identifies as "dust episodic pixels" only a fraction of the pixels used finally from NNMB for calculating dust DRE, then its use becomes not important for this study. So if someone trusts NNMB for DRE calculations, then it is much more easy to trust it also for dust outbreak identification.**

We think that our previous responses give a sufficient answer to the reviewer's summary comment.

**- There are more than 100 references and a lot of discussion about aerosol effects and model applications, but very few about NNMB validation on e.g. AOD retrievals. And only one (Ohmura) on BSRN radiation related validation. I think it is more essential to prove the validity of AOD NNMB output (e.g. radiation) and intermediate parameters (e.g. AOD), than a numerous studies cited here, with a very theoretical link to the paper.**

It is not the first time that NMMB is used, so validation of its AOD has already been done. In our paper, we have included all the available studies regarding the evaluation of the simulated AODs relied on the same NMMB version which is used here (Lines 294-306). Moreover, in the revised manuscript we are providing the weblink of the SDS-WAS System (https://sds-was.aemet.es/forecast-products/forecast-evaluation) in which is presented the forecast evaluation of NMMB AODs, among other aerosol models, utilizing ground-based (AERONET) and satellite (MODIS) retrievals as reference.

Concerning BSRN, we would like to remind and underline that it provides just reference radiation measurements. The BSRN is considered the best global network of quality radiation measurements. There is a very high number of scientific papers (http://bsrn.awi.de/other/publications/reviewed-scientific-papers-referring-to-bsrn/) or reports (http://bsrn.awi.de/other/publications/other-related-reports-and-papers/) referring to BSRN, so there is no need to make further reference to it than to the key paper of Ohmura et al. (1998) which is commonly used as reference for BSRN data. The validity of NMMB radiation fluxes is exactly proved through their comparison against BSRN measurements.

**- The validation using BSRN is incomplete. In the document and in the abstract you are talking about this validation and 8 stations. Then in the manuscript only one station is shown. And from that only 4 days. In order to validate the results a more comprehensive analysis of long**

**term periods of these 8 stations is needed. Probably Ohmura has answered some of the validation related questions, but this paper focuses on "intense dust outbreaks", and a specific model, so results might differ from the Ohmura related ones.**

We would like to point out that the calculated biases (NMMB-BSRN) over the hindcast periods, for each case and for each station (6 in total), are given already for the SW and LW radiation in Tables S2 and S3, respectively and discussion (lines 882-887) refers to their results. In the main text, we have decided to present just as an example the obtained results for the SW (first row in Figure 10) and LW (second row in Figure 10) radiation for two dust outbreaks (22/2 -25/2/2004 and 21/4-24/4/2007) that affected the Sede Boker station, for which concurrent AERONET retrievals were available. This allows us to give a better insight regarding the factors that can affect the level of agreement between model and ground observations. We agree with the reviewer that a long-term evaluation is valuable (i.e. identification of systematic errors) but for our purpose focus is given only on specific desert dust outbreaks trying to investigate if the inclusion of dust-radiation interaction in the numerical simulations can improve the forecasting skills of the NMMB-MONARCH model.

**- There are several issues that have to be clarified/commented on the input parameters of the model:**

**(i) Optical properties proposed in figure 2. Have been validated?**

The optical properties have not been validated. The model dust optical properties are based on single-particle optical properties derived by the GOCART model (Chin et al., 2002) and refractive indices from the Global Aerosol Data Set (GADS) (Koepke et al.,1997). Both datasets are very well known and very much often used and cited in literature, and therefore we believe that there is no need for further validation here.

**(ii) Water vapor, carbon dioxide, ozone, methane and oxygen. Where do you find these inputs?**

Water vapor comes from the model simulations. We used a fixed value of $CO_2$ (350 ppm), methane (1.5 ppm) and oxygen; and a seasonal climatology for ozone.

**(iii) Differences in dust optical properties of Sahara and middle East sources. What did you use and how much uncertain are they? and what is the contribution of this uncertainty in the final DRE budget?**

The dust single-particle optical properties and the emitted size distribution are constant throughout the simulation domain without discriminating between different dust sources (Sahara, Middle East). At each forecast step, the aerosol optical depth (AOD), the single scattering albedo (SSA) and the asymmetry parameter (ASYM) have been produced based on the formulas presented in Pérez et al. (2006) utilizing the simulated mass concentration, the GOCART single-particle optical properties and the refractive indices from the Global Aerosol Data Set (GADS) which have been modified according to Sinyuk et al. (2003), as it has been described in Pérez et al. (2011) (lines 331-336). Regarding the last question of the reviewer, in order to be give an accurate answer a sensitivity analysis is required. More specifically, it must be investigated how the variation of key aerosol optical properties (AOD, SSA and ASYM) will affect the perturbations of the radiation budget and subsequently the associated impacts on dust AOD, dust emission, meteorological variables and radiation. This is something that has not been done in the present paper but it will be considered in a future work dedicated to all the aforementioned aspects considering also other parameters (e.g., dust layer vertical extension) which can affect DREs.

**- BSRN and model differences in wavelength integrals of solar radiation. You mention: "These differences might contribute to the level of agreement between model and observations;**

**however, are not discussed in our evaluation analysis".** **I think this is an important issue that have to be clearly discussed if a proper validation is included.**

For solar radiation, the NMMB-BSRN SW flux departures, attributed to the different spectral coverage and integrals, are minor, varying from 1 to1.5 % (higher values for the model), therefore they do not affect substantially the agreement (in terms of biases) between model and measured fluxes.

**- As already mentioned AOD comparisons from MODIS and NNMB could add value to this work.** **"The model's ability to reproduce correctly the spatial patterns and values of dust AODs is crucial for a successful computation of the dust DREs, since DREs are determined to a large extent by AOD". In addition you are mentioning modis uncertainty in section 2. Is this getting high (e.g. ~ 0.5) for both sea and mostly surface retrievals when you examine AODs in the order of 2-3 based on the table 1?** **And is this uncertainty already important for such outbreaks for the GM and indirectly for the DRE related uncertainty?**

Actually, the uncertainty of C051 MODIS AOD retrievals is not reported in section 2, where only the detection of dust outbreaks is described. The uncertainty of MODIS AOD retrievals over ocean is ±0.03±0.05*AOD (Remer et al., 2002) while over land is higher and equal to ±0.05±0.15*AOD (Levy et al., 2010). The maximum MODIS retrieved AOD, over both continental and maritime areas, do not exceed 5, which means that the AOD uncertainties above sea and land, in absolute terms, are smaller than 0.28 and 0.8, respectively. In our cases, but also in general, these maximum AOD uncertainties are locally restricted and not recorded frequently (see Figures 3 and S1) while uncertainties are generally smaller, and thus do not affect the GM. Moreover, they neither affect DREs, since as already explained in our previous responses and in the manuscript, the DREs have been computed via the NMMB simulations without setting any constrain depending on MODIS retrievals (i.e., availability, magnitude).

**- Table 2. These statistics are not referring to the model uncertainty but is an averaging of the episodes provided by the GM. NNMB DRE uncertainty is much more useful for any future user of these results. For example a systematic bias can not be identified here. This is also because the GM thresholds are mostly subjective as:**

**(i) Mean AOD values on dust related areas do not have an important statistical meaning due to the non normal distribution of AOD. It is clear that this is a published work and I have tried to follow the previous work by Gkikas et al and the relative open discussion, describing the method. However, as this is an open public statement I have to comment that AOD does not follow necessarily a normal distribution so using the mean is not absolutely correct. Moreover, dust outbreaks related pixels/locations can be characterized more from a bimodal distribution of AODs when another (than dust) important AOD source is rarely present (e.g. most of the marine grids of Mediterranean domain).**

First of all, as stated by the Referee, we would like to remind that the GM method has already been published (Gkikas et al., 2013; 2016) just after the discussion that took place concerning the way of computation of AOD thresholds, i.e. geometric versus arithmetic mean AOD values, which implies its validity against similar arguments cited in this comment. Nevertheless, we can remind the following. We agree with the Reviewer that AOD follows a log-normal rather than a Gaussian distribution, and that the arithmetic mean and standard deviation are not probably the best metrics for the calculation of the AOD thresholds, even though both primary statistics are widely applied in numerous aerosol studies. During the review process of Gkikas et al. (2013), following a similar comment raised by one of the referees, proposing to calculate the AOD thresholds based on the

geometric mean and geometric standard deviation, we recomputed the AOD thresholds and compared them to the typical ones already used (based on arithmetic mean and standard deviation). Although there were found some differences in the thresholds' magnitude, in general, the geographical patterns of AOD thresholds were similar for both strong and extreme DD episodes. As for strong episodes, those differences were rather small, for example typical AOD thresholds varied within the range 0.4-1.2 and the geometrical thresholds ranged from 0.4 to 1.6. On the other hand, larger differences existed for extreme DD episodes, with the typical thresholds ranging from 0.6 to 2.2 while the geometric ones varying from 1 to more than 10. However, such extremely high AOD values are extremely rare and using them would be unrealistic from the physical point of view. For these reasons, it was decided to rely on GM methodology of Gkikas et al. (2013).

**(ii) GM: By definition high mean AOD values per pixel are closer to dust sources. That makes possible that a pixel with high (in an absolute sense) AOD close to a dust source to be considered non episodic and a pixel with lower AOD, away from the sources to be considered episodic. This is ok, as it is just a matter of definition. But it gets more important when it is used for DRE calculations. So, the latest can be problematic when you calculate DRE in dust outbreaks or filter the outbreaks, as for the first example pixel (high AOD) it is not an outbreak and for the second (lower AOD) it is characterized as an outbreak. The results using this method for DRE calculations become not easily useful and applicable.**

The issue of the identification method of DD outbreaks based on pixel-level AOD values, has already been addressed in our previous papers using the GM methodology, following similar comments to the one made by the Referee here. It has been shown that any differences in terms of AOD thresholds and dust outbreaks features (frequency, intensity) were not substantial.

However, the most important concerning the rest of Referee's comment referring to possible effects of this issue on computed DREs here, we would like to clarify again that DREs are computed by NMMB and have nothing to do with the AOD thresholds. It should be clear and kept in mind that GM methodology is only used for the determination of days with intense dust outbreaks for which NMMB then operates and makes computations of DREs all over the domain.

**- Last but very important, the paper is very long and in various cases the discussion includes a lot of details that in the end confuse the reader on what is the important findings here and which are not. Even for scientists in the field it becomes difficult to read. Authors have to try to reduce the length of the manuscript keeping the important aspects of the results presented. Basically for section 5 I would suggest to try to take out a lot of information that are secondary and to focus on the important results.**

In the revised manuscript, following the suggestion of the Reviewer, we made an effort and reduced the paper length by removing some parts which can be considered as secondary information. However, at the same time, also following the Reviewers' suggestions, we added a discussion about the quantitative intercomparison between MODIS and NMMB as well as about the potential improvements on short-term forecasts of the temperature fields by the model. Therefore, the final length of the revised manuscript is similar to that of the original manuscript. We believe that any further shortening of the manuscript would be at the expense of its quality and scientific content.

**Minor comments:**

**Line 141: it has already mentioned previously.**

It has been modified.

**Line 173: developed – improved**

Done.

**Table 1: episodes = grid cells**

We think that is already clearly stated in the caption.

**The overall approach of this paper is valuable and worth publishing. I strongly believe that after the above revisions, corrections and additional analysis it will be essentially upgraded and then it could be published in the ACP journal.**